# HIV-1 Vpr antagonizes innate immune activation by targeting karyopherin-mediated NF-κB/IRF3 nuclear transport

Hataf Khan[†‡], Rebecca P Sumner[†], Jane Rasaiyaah[§], Choon Ping Tan[#], Maria Teresa Rodriguez-Plata[¶], Chris Van Tulleken, Douglas Fink, Lorena Zuliani-Alvarez, Lucy Thorne, David Stirling[**], Richard SB Milne, Greg J Towers[*]

Division of Infection and Immunity, University College London, London, United Kingdom

**\*For correspondence:**
g.towers@ucl.ac.uk

[†]These authors also contributed equally to this work

**Present address:** [‡]Department of Infectious Diseases, King's College London, London, United Kingdom; [§]Molecular and Cellular Immunology Unit, UCL Great Ormond Street Institute of Child Health, London, United Kingdom; [#]Translation & Innovation Hub, London, United Kingdom; [¶]Black Belt TX Ltd, Stevenage Bioscience Catalyst, Stevenage, United Kingdom; [**]Broad Institute of MIT and Harvard University, Cambridge, United States

**Abstract** HIV-1 must replicate in cells that are equipped to defend themselves from infection through intracellular innate immune systems. HIV-1 evades innate immune sensing through encapsidated DNA synthesis and encodes accessory genes that antagonize specific antiviral effectors. Here, we show that both particle associated, and expressed HIV-1 Vpr, antagonize the stimulatory effect of a variety of pathogen associated molecular patterns by inhibiting IRF3 and NF-κB nuclear transport. Phosphorylation of IRF3 at S396, but not S386, was also inhibited. We propose that, rather than promoting HIV-1 nuclear import, Vpr interacts with karyopherins to disturb their import of IRF3 and NF-κB to promote replication in macrophages. Concordantly, we demonstrate Vpr-dependent rescue of HIV-1 replication in human macrophages from inhibition by cGAMP, the product of activated cGAS. We propose a model that unifies Vpr manipulation of nuclear import and inhibition of innate immune activation to promote HIV-1 replication and transmission.

## Introduction

Like all viruses, lentiviruses must navigate the hostile environment of the host cell in order to infect, produce new viral particles, and transmit to new cells. A principal feature of cellular defences is detection or sensing of incoming viruses and subsequent production of inflammatory cytokines, particularly type one interferons (IFNs). All viral infections have the potential to trigger IFN *in vivo* through viral pathogen associated molecular patterns (PAMPs) activating pattern recognition receptors (PRR). The degree to which each virus does this, and their capacity to antagonize IFN activity and its complex effects, are key in determining transmission mechanism, host range, and disease pathogenesis. Like other viruses, lentiviruses also antagonize specific host proteins or pathways that would otherwise suppress infection. Lentiviruses typically do this through accessory gene function. For example, HIV-1 antagonizes IFN-induced restriction factors through accessory genes encoding Vif (APOBEC3G/H), Vpu (tetherin), and Nef (tetherin/SERINC3/5) reviewed in *Foster et al., 2017*; *Sumner et al., 2017*.

The HIV-1 accessory protein Vpr interacts with and manipulates many proteins including its cofactor DCAF1 (*Zhang et al., 2001*), karyopherin alpha 1 (KPNA1, importin α) (*Miyatake et al., 2016*), the host enzyme UNG2 (*Wu et al., 2016*) as well as HTLF (*Lahouassa et al., 2016*; *Yan et al., 2019*), SLX4 (*Laguette et al., 2014*), and CCDC137 (*Zhang and Bieniasz, 2020*). Indeed, Vpr has been shown to significantly change infected cell protein profiles, affecting the level of hundreds of proteins in proteomic studies, likely indirectly in most cases, consistent with manipulation of central mechanisms in cell biology (*Greenwood et al., 2019*). Vpr has also been shown to both enhance

(*Liu et al., 2014*; *Liu et al., 2013*; *Vermeire et al., 2016*) or decrease NF-κB activation (*Harman et al., 2015*; *Trotard et al., 2016*) in different contexts and act as a cofactor for HIV-1 nuclear entry, particularly in macrophages (*Vodicka et al., 1998*). However, despite this work, the mechanistic details of Vpr promotion of HIV replication are poorly understood and many studies seem contradictory. This is partly because the mechanisms of Vpr-dependent enhancement of HIV-1 replication are context dependent, and cell type specific, although most studies agree that Vpr is more important for replication in macrophages than in T cells or PBMC (*Connor et al., 1995*; *Dedera et al., 1989*; *Fouchier et al., 1998*; *Hattori et al., 1990*; *Mashiba et al., 2015*). Manipulation of host innate immune mechanisms by Vpr to facilitate replication in macrophages has been suggested by various studies, although there has been no clear mechanistic model or understanding how particular Vpr target proteins link to innate immune manipulation (*Harman et al., 2015*; *Liu et al., 2014*; *Okumura et al., 2008*; *Trotard et al., 2016*; *Vermeire et al., 2016*).

Many viruses have been shown to manipulate innate immune activation by targeting transcription factor nuclear entry downstream of PRR. For example, Japanese encephalitis virus NS5 targets KPNA2, 3, and 4 to prevent IRF3 and NF-KB nuclear translocation (*Ye et al., 2017*). Hantaan virus nucleocapsid protein inhibits NF-KB p65 translocation by targeting KPNA1, -2, and -4 (*Taylor et al., 2009*). Most recently, vaccinia virus protein A55 was shown to interact with KPNA2 to disturb its interaction with NF-KB (*Pallett et al., 2019*). Hepatitis C virus NS3/4A protein restricts IRF3 and NF-κB translocation by cleaving KPNB1 (importin-β) (*Gagné et al., 2017*).

HIV-1 Vpr has also been linked to Karyopherins and manipulation of nuclear import. Vpr has been shown to interact with a variety of mouse (*Miyatake et al., 2016*), yeast (*Vodicka et al., 1998*) and human karyopherin proteins including human KPNA1, 2, and 5 (*Nitahara-Kasahara et al., 2007*). Indeed, the structure of a C-terminal Vpr peptide (residues 85–96) has been solved in complex with mouse importin α2 (*Miyatake et al., 2016*). Here, we demonstrate that Vpr inhibits innate immune activation downstream of a variety of viral and non-viral PAMPs by inhibiting nuclear transport of IRF3 and NF-KB by KPNA1. We confirm Vpr interaction with KPNA1 by co-immunoprecipitation and link Karyopherin binding and inhibition of innate immunity by showing that Vpr prevents interaction between KPNA1 and IRF3/NF-KB *in vitro*. Critically, we show that Vpr (F34I/P35N) fails to inhibit nuclear transport of IRF3 and NF-KB, fails to antagonize innate immune sensing, and fails to interact with KPNA1. We demonstrate that Vpr mutants that do not recruit to the nuclear envelope cannot antagonize innate sensing but retain induction of cell cycle arrest, genetically separating key Vpr functions. Importantly, by targeting activated transcription factors, Vpr prevents innate immune activation by a wide range of non-viral agonists suggesting Vpr has roles beyond inhibiting innate immune activation of PAMPs derived from the virus itself. Our new findings support a unifying model of Vpr function, consistent with much of the Vpr literature, in which Vpr associated with incoming viral particles suppresses nuclear entry of activated inflammatory transcription factors to facilitate HIV-1 replication in innate immune activated macrophages.

## Results

### HIV-1 replication in cGAMP-stimulated MDMs requires Vpr

A considerable body of evidence suggests an important role for Vpr in supporting HIV-1 replication in macrophages, but the relevant Vpr mechanisms for this function have been enigmatic. We set out to investigate the role of Vpr in manipulating host innate immune mechanisms during HIV-1 infection of primary human cells. We prepared human monocyte-derived macrophages (MDM) by purifying monocytes from peripheral blood by adherence and treating with M-CSF (*Rasaiyaah et al., 2013*). Macrophages prepared in this way are particularly permissive to HIV-1 replication facilitating study of HIV-1 biology in a primary myeloid cell type. We found that wild-type HIV-1 and HIV-1ΔVpr replicated equally well in (MDM) (*Figure 1A*; *Rasaiyaah et al., 2013*) Consistent with previous studies, wild-type HIV-1, and HIV-1 deleted for Vpr replicated equally well in activated primary human CD4+ T cells (*Figure 1—figure supplement 1A*; *Dedera et al., 1989*; *Fouchier et al., 1998*).

Vpr has been shown to antagonize innate immune signaling in HeLa cells reconstituted for DNA sensing by STING expression (*Trotard et al., 2016*), so we hypothesized that Vpr might be particularly important when DNA sensing is activated. To test this, we mimicked activation of the DNA sensor cGAS by treating MDM with cGAMP, the product of activated cGAS. In the presence of cGAMP,

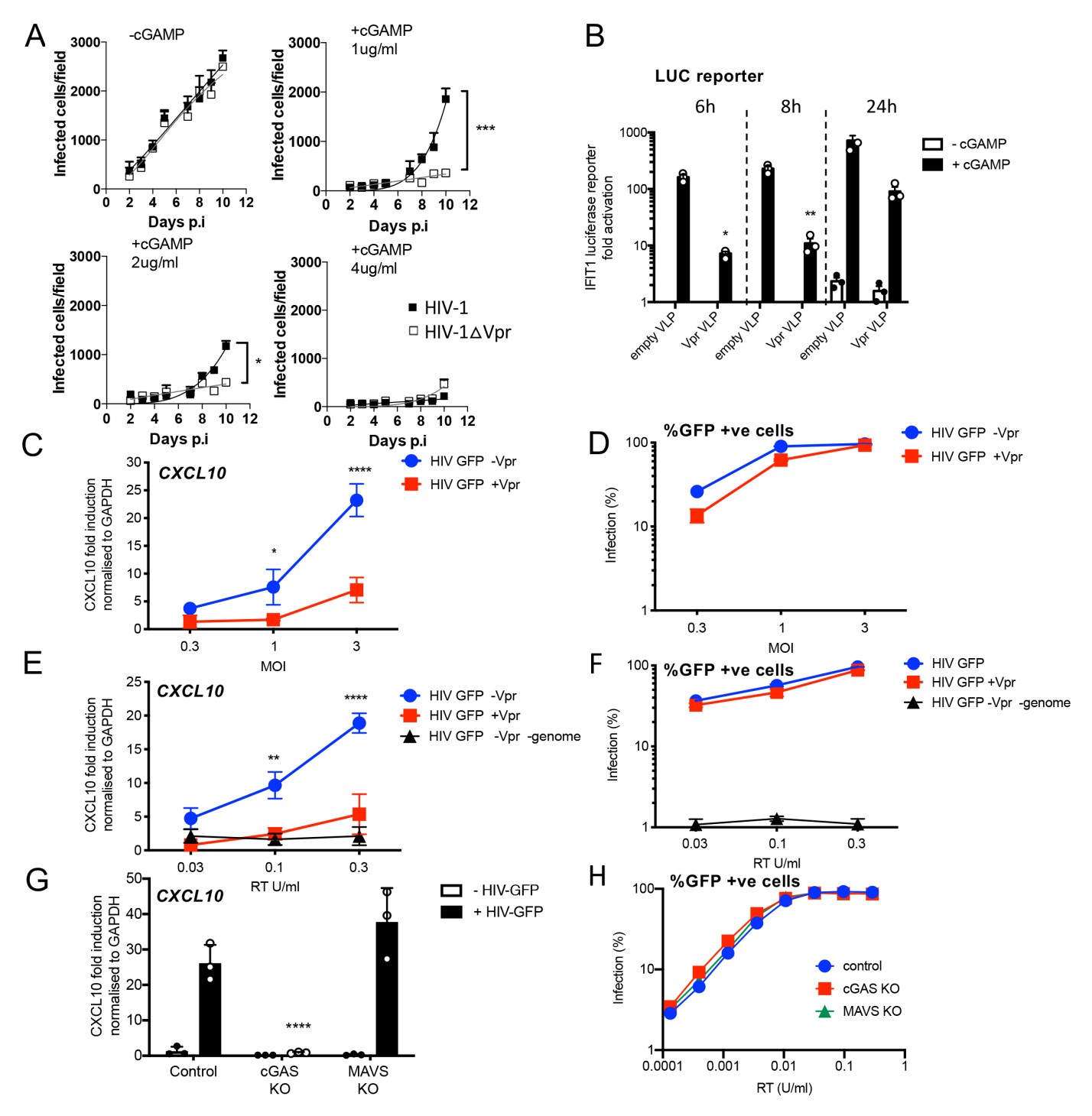

**Figure 1.** HIV-1 replication in cGAMP-stimulated MDMs requires Vpr. (**A**) Replication of WT Yu2 HIV-1 or Yu2 HIV-1ΔVpr in MDMs stimulated with 1 µg/ml, 2 µg/ml or 4 µg/ml cGAMP or left unstimulated, infection measured by counting Gag-positive cells stained with anti-p24. Mean+/-SEM n = 3 1 and 2 µg/ml cGAMP; n = 2 4 µg/ml cGAMP. *** = two-way ANOVA p value < 0.001, *=p < 0.05. (**B**) Fold induction of IFIT1-Luc after activation of STING by cGAMP (5 µg/ml) and infection with HIV-1 virus-like particles (VLP) lacking genome and bearing Vpr (+Vpr) or lacking Vpr (-Vpr) (1 RT U/ml) in IFIT1-Luc reporter THP-1 cells. cGAMP and virus were added to cells at the same time. (**C**) Fold induction of *CXCL10* after infection of THP-1 cells with HIV-GFP -Vpr or HIV-GFP +Vpr at the indicated MOI. (**D**) Percentage of THP-1 cells infected by HIV-GFP -Vpr or HIV-GFP +Vpr in (**C**). (**E**) Fold induction of *CXCL10* after infection of THP-1 cells with HIV-GFP -Vpr, HIV-GFP +Vpr, or HIV-1 particles lacking Vpr and genome, at indicated doses measured by reverse transcriptase SG-PERT assay. (**F**) Percentage of THP-1 cells infected by HIV-GFP viruses in (**E**). (**G**) Fold induction of *CXCL10* after infection of unmodified control, cGAS-/-or MAVS-/- THP-1 knock out cells with HIV-GFP lacking Vpr (0.3 RT U/ml). (**H**) Percentage infection of control, cGAS-/-or

*Figure 1 continued on next page*

*Figure 1 continued*

MAVS-/- THP-1 knock out cells infected with HIV-GFP at indicated doses of RT (SG-PERT). (B–H) Data are expressed as means ± SD (n = 3) with two-way ANOVA * (p<0.05), ** (p<0.01), *** (p<0.001), **** (p<0.0001) compared to virus without genome (B), HIV GFP+Vpr (C, E) and control (G).

The online version of this article includes the following figure supplement(s) for figure 1:

**Figure supplement 1.** HIV-1 replication in cGAMP-stimulated MDMs requires Vpr and Vpr suppresses HIV-1 innate immune sensing by cGAS.

HIV-1 replication in MDM was, indeed, Vpr-dependent. One µg/ml cGAMP specifically suppressed HIV-1ΔVpr more potently than wild-type virus and 4 µg/ml cGAMP overcame Vpr activity and suppressed replication of both wild-type and mutant viruses (*Figure 1A*). Intriguingly, Vpr did not rescue HIV-1 replication from cGAMP-mediated inhibition in primary human CD4+ T cells, and cGAMP had only minimal effect on HIV-1 replication in Jurkat T cells (*Figure 1—figure supplement 1A*). These data demonstrate that HIV-1 replication in cGAMP-stimulated MDM is Vpr-dependent. They are consistent with previous observations suggesting Vpr is more important in macrophages than T cells and that the consequences of cGAMP treatment differ between these cell types (*Gulen et al., 2017*; *Xu et al., 2016*).

## HIV-1 particle delivered Vpr inhibits gene expression stimulated by DNA sensing

We next investigated the effect of particle-associated Vpr on innate immune activation. The myeloid cell line THP-1 expresses cGAS and STING and has a functional DNA-sensing pathway (*Mankan et al., 2014*). We used THP-1 cells expressing the Gaussia luciferase gene under the control of the endogenous *IFIT1* promoter (herein referred to as THP-1 IFIT1-luc) (*Mankan et al., 2014*) to measure the effect of Vpr on cGAMP-induced IFIT1-luc expression. IFIT1 (ISG56) is a well-characterized ISG that is highly sensitive to cGAMP and type 1 IFN. Treatment of THP-1 IFIT-luc cells with cGAMP induced IFIT1-luc expression by two orders of magnitude. This activation was significantly suppressed if cells were infected with VSV-G pseudotyped, genome-free, HIV-particles bearing Vpr (referred to here as virus-like particles or VLP), but not by VLP lacking Vpr, immediately prior to cGAMP addition (*Figure 1B*). IFIT1-Luc was measured 6, 8, and 24 hr after cGAMP addition/infection.

In this experiment, doses of VLP required to suppress IFIT1-luc expression were high, equivalent to a multiplicity of infection of 20 as measured by correlating VLP reverse transcriptase levels (SG-PERT) (*Vermeire et al., 2012*), with HIV-1 GFP titers on THP-1. We assume that such a high dose of Vpr-bearing VLP is required because cGAMP treatment activates numerous STING complexes in most of the cGAMP-treated cells. If this effect of Vpr is relevant to infection, we expect that cGAS/STING activated by the incoming HIV genome should be sensitive to the amount of Vpr contained in an individual particle. To test this, we activated DNA sensing using high-dose infection by VSV-G pseudotyped HIV-1 vectors bearing GFP-encoding genome. We used an HIV-1 packaging plasmid, derived from HIV-1 clone R9, encoding Gag-Pol, Tat and Rev (p8.91) or Gag-Pol, Tat and Rev and Vpr, Vpu, Vif and Nef (p8.2) (*Zufferey et al., 1997*). Strikingly, although Vpr-positive and negative HIV-1 GFP stocks infected THP-1 cells to similar levels (*Figure 1D*), induction of inflammatory cytokine, and ISG, CXCL10 was reduced if the HIV-1 GFP carried Vpr (*Figure 1C*). This indicates that Vpr can inhibit the consequences of sensing driven by the Vpr bearing virus particles themselves.

Genome-free, non-infectious, HIV-1 particles did not induce CXCL10 expression (*Figure 1E,F*), evidencing the importance of viral DNA in this response. Furthermore, CXCL10 expression was not induced after infection of THP-1 cGAS knock out cells, consistent with CXCL10 induction being cGAS-dependent (*Figure 1G*). Knock out of the RNA-sensing adaptor protein MAVS had no effect on induction of CXCL10 (*Figure 1G*). cGAS and MAVS knock out were confirmed by immunoblot (*Figure 1—figure supplement 1C*).

As expected, a lower dose of virus was required to see the effect of Vpr when the particles themselves activated sensing, and in this latter experiment, Vpr effects were clear at MOIs of 3 (*Figure 1C,E*). Moreover, single round titer of HIV-1 GFP was not affected by cGAS or MAVS knock out, confirming that sensing activation does not impact single round infectivity of HIV-1 GFP VSV-G pseudotypes in this assay consistent with HIV-1 vector not being particularly sensitive to IFN (*Figure 1H*, *Figure 1—figure supplement 1B*).

## HIV-1 Vpr expression inhibits innate immune activation

We next tested whether Vpr expressed in isolation can suppress innate immune activation by cGAMP. Vpr from the primary founder HIV-1 clone SUMA (*Fischer et al., 2010*) was expressed in THP-1 IFIT1-luc cells using an HIV-1 vector we called pCSVIG (*Figure 2—figure supplement 1A*, B). Vpr was expressed using MOIs of approximately 0.2–1. Forty hours after transduction, cells were treated with cGAMP (5 µg/ml), and IFIT1-luc was measured 8 hr later. Prior expression of Vpr reduced IFIT1-luc responses in a dose-dependent manner, whereas the highest dose of empty vector had no effect, measured as a negative control (*Figure 2A*; infection data in *Figure 2—figure supplement 1C*). Vpr expression (MOI = 1, *Figure 2—figure supplement 1D*) also suppressed cGAMP-mediated induction of endogenous ISG mRNA expression, measured by qRT-PCR for *MxA*, *CXCL10*, *IFIT2*, and *viperin* (*Figure 2B*) and inhibited cGAMP-induced CXCL10 secretion (*Figure 2C*; infection data to gauge MOI in *Figure 2—figure supplement 1E*).

IFIT1-luc expression stimulated by transfection of herring testis (HT) DNA was also inhibited by Vpr expression, consistent with the notion that Vpr antagonizes DNA sensing (*Figure 2D*, *Figure 2—figure supplement 1F*). Strikingly, Vpr also reduced Sendai-virus-induced activation of IFIT1-luc, which is mediated by MDA5 and RIGI RNA sensing (*Andrejeva et al., 2004*; *Rehwinkel et al., 2010*; *Figure 2E*, *Figure 2—figure supplement 1G*) and IFIT1-luc activation after stimulation with the TLR4 ligand LPS (*Figure 2F*, *Figure 2—figure supplement 1H*). Thus, Vpr expression appeared to mediate a generalized suppression of innate immune activation.

## Vpr inhibition of innate immune activation is dependent on DCAF1 but independent of cell cycle arrest

In order to separate innate immune antagonism from other Vpr functions, we used three Vpr mutants with distinct functional deficits. Vpr R80A, is defective in inducing cell cycle arrest *Laguette et al., 2014*; Vpr Q65R fails to recruit DCAF1 and so cannot degrade target proteins *Laguette et al., 2014*; and Vpr F34I/P35N fails to bind cyclophilin A and does not localize to the nuclear membrane (*Vodicka et al., 1998*; *Zander et al., 2003*).

All three mutant Vprs were efficiently incorporated into HIV-1 GFP particles (*Figure 3A*). When delivered by viral particles, Vpr R80A effectively suppressed IFIT1-luc induction by cGAMP in THP-1 cells; however, Vpr Q65R and Vpr F34I/P35N had little if any suppressive effect (*Figure 3B*). In these experiments, cGAMP was added to the target cells directly after the virus. Suppression of IFIT1-luc induction by Vpr R80A suggested that cell cycle arrest was not required for innate immune antagonism. To further test this, we measured the effect of all three Vpr mutants on cell cycle progression. As reported, WT Vpr expression in THP-1 cells induced a significant increase of cells in G2/M phase of cell cycle and Vpr R80A had no effect (*Figure 3C*, *Figure 3—figure supplement 1G*; *Laguette et al., 2014*). Vpr F34I/P35N, which cannot effectively suppress cGAMP-mediated IFIT1-luc/ISG expression (*Figure 3B and G*), also induced G1/M cell cycle arrest, albeit slightly less efficiently than wild-type Vpr protein, as previously described (*Vodicka et al., 1998*; *Figure 3C*). The DCAF1 Vpr-binding mutant Q65R did not inhibit cell cycle, as reported (*Figure 3C*; *Laguette et al., 2014*). These data genetically separate the effects of Vpr expression on cell cycle, and on inhibition of innate immune activation, suggesting that these functions depend on manipulation of different target proteins. It is striking that amino acids at positions 34/35 and 80 are close in Vpr structures and distant from the UNG2-binding site, suggesting an additional target binding interface, as seen in the highly related Vpx protein (*Figure 3—figure supplement 1B,C*; *Morellet et al., 2003*; *Schwefel et al., 2014*; *Wu et al., 2016*).

We next asked whether DCAF-1 was required for innate immune antagonism, as suggested by the Vpr Q65R mutant, which fails to recruit DCAF1, and cannot suppress cGAMP-induced IFIT1-luc expression (*Figure 3B*). Depletion of DCAF1 in THP-1 cells by shRNA prevented Vpr from inhibiting cGAMP induction of IFIT1-luc (*Figure 3D*). Neither DCAF1 depletion, nor cGAMP treatment reduced infectivity of HIV-1 GFP vector (*Figure 3—figure supplement 1A*). Vpr was active in cells expressing a non-targeting shRNA (shControl) and suppressed IFIT1-luc induction (*Figure 3D*). Expression of empty (no Vpr) vector had no effect on IFIT1-luc induction (*Figure 3D*). Effective depletion of DCAF1 was evidenced by immunoblot (*Figure 3E*). Thus, Vpr inhibition of innate immune activation requires DCAF1.

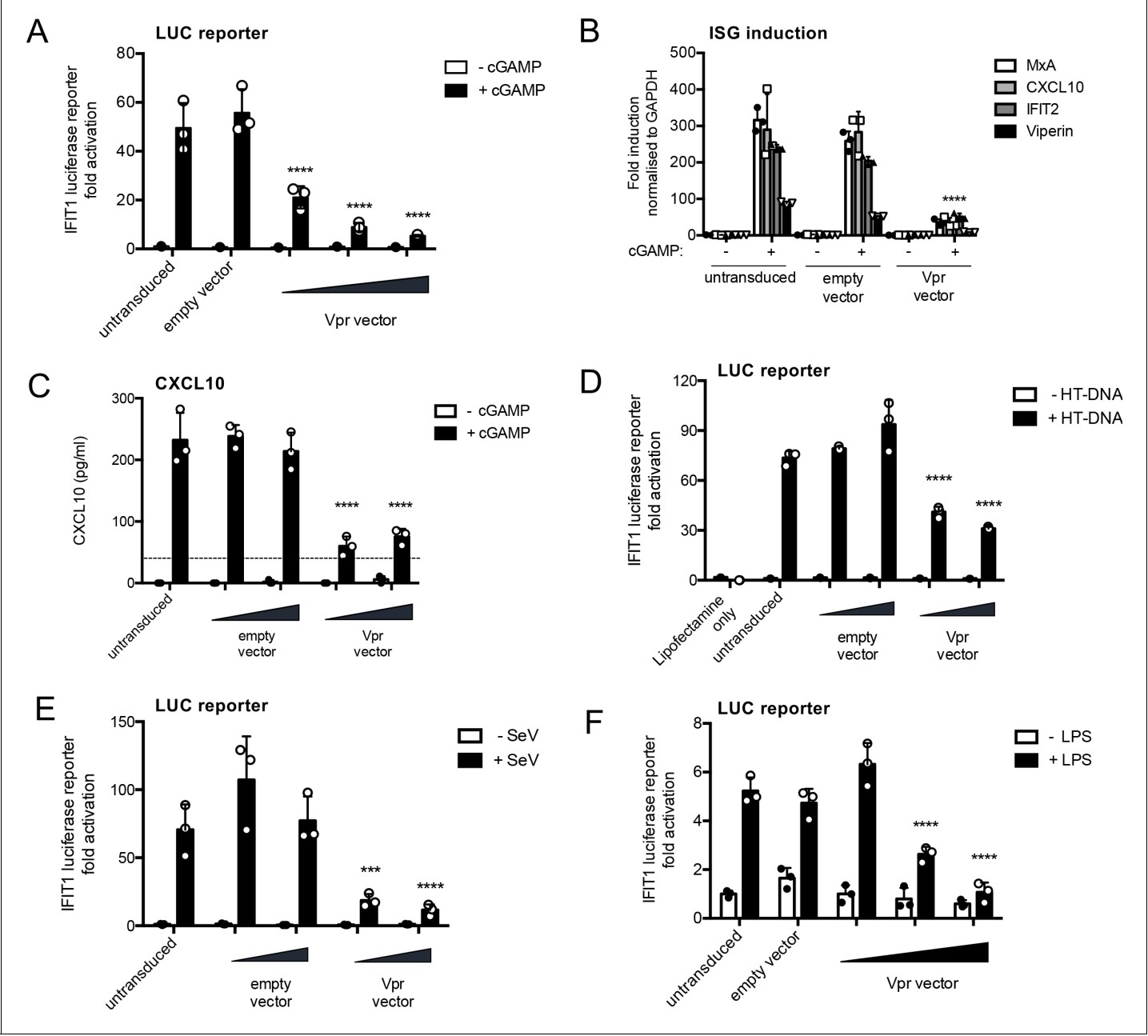

**Figure 2.** HIV-1 Vpr expression inhibits interferon-stimulated gene expression after stimulation with various innate immune stimuli. (**A**) Fold induction of IFIT1-Luc, after activation of STING by cGAMP (5 μg/ml), in IFIT1-Luc reporter THP-1 cells expressing Vpr from a lentiviral vector delivered at MOIs of 0.25, 0.5, 1, or after empty vector transduction (MOI 1) or in untransduced cells. (**B**) Fold induction of ISGs MxA, CXCL10, IFIT2, and Viperin after activation of STING by cGAMP (5 μg/ml) in cells expressing Vpr from a lentiviral vector (MOI 1), or after empty vector transduction (MOI 1) or in untransduced THP-1 cells. (**C**) Secreted CXCL10 (ELISA) after activation of STING by cGAMP (5 μg/ml) in cells expressing Vpr from a lentiviral vector (MOI 0.5, 1), or after transduction with empty vector (MOI 0.5, 1) or in untransduced THP-1 cells. Dotted line shows limit of detection. (**D**) Fold induction of IFIT1-Luc after HT-DNA transfection (5 μg/ml) of cells expressing Vpr from a lentiviral vector (MOI 0.5, 1), or empty vector (MOI 0.5, 1) or in untransduced IFIT1-Luc reporter THP-1 cells. (**E**) Fold induction of IFIT1-Luc, after Sendai virus infection, of cells expressing Vpr from a lentiviral vector (MOI 0.5, 1), or after transduction by empty vector (MOI 0.5, 1) or in untransduced IFIT1-Luc reporter THP-1 cells. (**F**) Fold induction of IFIT1-Luc, after LPS treatment (1 μg/ml), of cells expressing Vpr from a lentiviral vector (MOI 0.25, 0.5, 1), after transduction by empty vector (MOI 1) or in untransduced IFIT1-Luc reporter THP-1 cells. Data are expressed as mean ± SD (n = 3) analyzed using two-way ANOVA * ($p < 0.05$), ** ($p < 0.01$), *** ($p < 0.001$), **** ($p < 0.0001$) compared to data for empty vector. n = 3 (**A**, **D–F**) or 2 (**B–C**) independent experiments.

The online version of this article includes the following figure supplement(s) for figure 2:

**Figure supplement 1.** HIV-1 Vpr expression inhibits interferon-stimulated gene expression after stimulation with various innate immune stimuli.

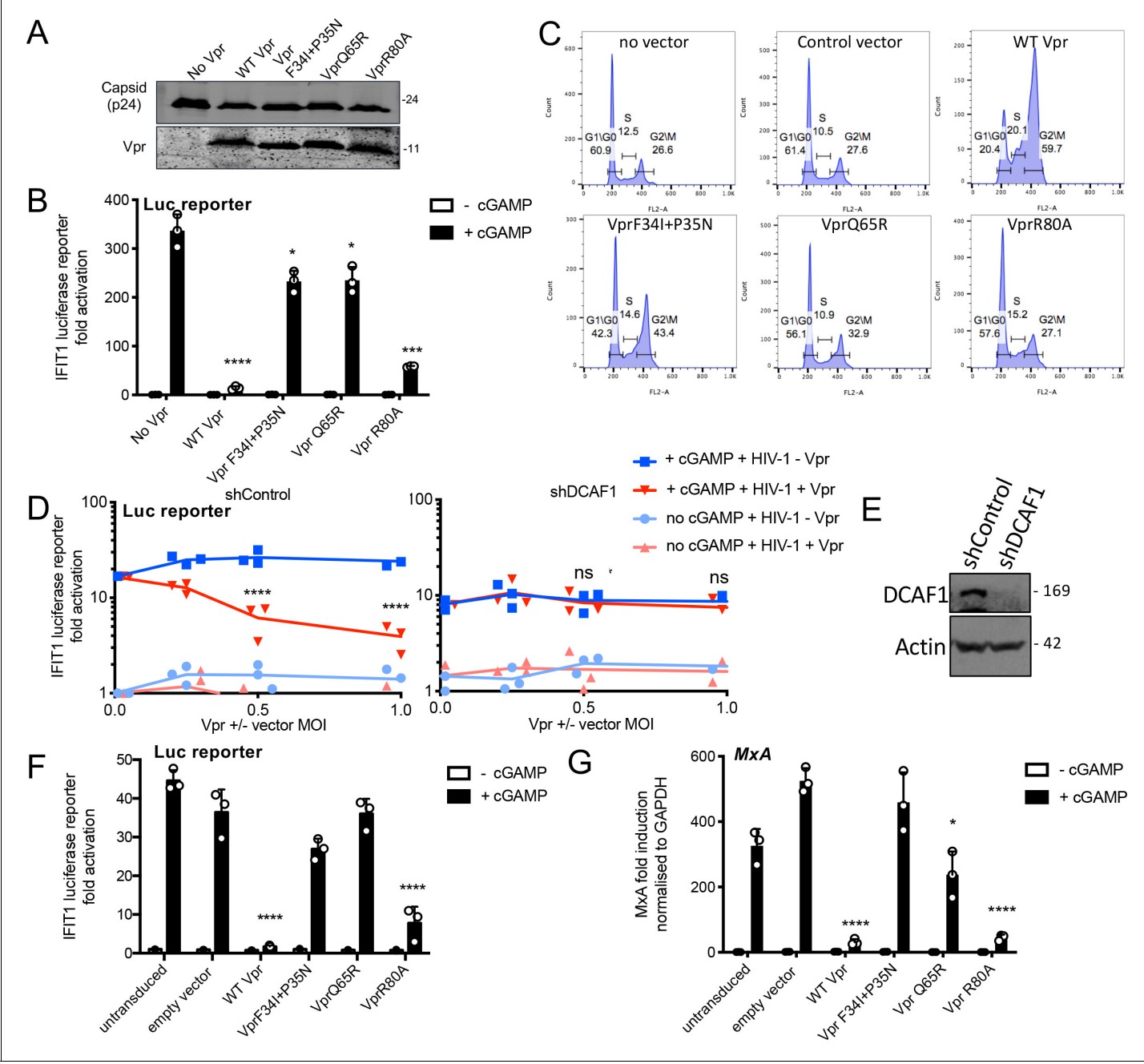

**Figure 3.** Vpr inhibition of innate immune activation is dependent on DCAF1 but independent of cell cycle arrest. (**A**) Immunoblot detecting p24 (capsid) and Vpr in pelleted VSV-G pseudotyped VLP lacking genome used in (**B**). Size markers in kDa are indicated on the right. (**B**) Fold induction of IFIT1-Luc after activation of STING by cGAMP (5 µg/ml) and infection with VLP bearing WT or mutant Vpr, or lacking Vpr (1 RT U/ml) in IFIT1-Luc reporter THP-1 cells. Cells were infected at the same time as cGAMP treatment. (**C**) Flow cytometry plots showing cell cycle phases of THP-1 cells transduced with an empty vector, WT Vpr, or mutant Vpr, encoding vector (MOI 1) or left untransduced as a control and stained with propidium iodide to label DNA. Percentage cells in each cell cycle stage are shown. (**D**) Fold induction of IFIT1-Luc after activation of STING by cGAMP (5 µg/ml) in cells expressing Vpr from a lentiviral vector, or expressing empty vector, or in untransduced IFIT1-Luc reporter THP-1 cells expressing a control, or a DCAF1 targeting shRNA. Mean +/- SEM n = 3 independent experiments. (**E**) Immunoblot detecting DCAF1, or actin as a loading control, from extracted THP-1 cells expressing a non-targeting, or DCAF1-targeting, shRNA. Size markers are shown in kDa on the right. (**F**) Fold induction of IFIT1-Luc after activation of STING by cGAMP (5 µg/ml) in cells expressing WT, or mutant, Vpr from a lentiviral vector (MOI 1), or empty vector (MOI 1) or in untransduced IFIT1-Luc reporter THP-1 cells. (**G**) Fold induction of MxA mRNA after activation of STING by cGAMP (5 µg/ml) in cells expressing WT, or mutant, Vpr from a lentiviral vector (MOI 1), or after transduction by empty vector (MOI 1) or in untransduced THP-1 cells. Data are mean ± SD (n = 3). Two-way ANOVA

*Figure 3 continued on next page*

*Figure 3 continued*

test: * (p<0.05), ** (p<0.01), *** (p<0.001), **** (p<0.0001) compared to no Vpr or empty vector controls. Data are representative of three (**B–D, F**) or two (**A, E, G**) independent experiments.

The online version of this article includes the following figure supplement(s) for figure 3:

**Figure supplement 1.** Vpr inhibition of innate immune activation is dependent on DCAF1 but independent of cell cycle arrest.

Expressed Vpr had similar mutation sensitivity as Vpr delivered by HIV-1 particles (compare *Figure 3F,G and B*). Expression of wild-type Vpr, or Vpr R80A, prevented cGAMP activation of the IFIT1-luc reporter (*Figure 3F*), and induction of endogenous *MxA* message in THP-1 cells (*Figure 3G*, *Figure 3—figure supplement 1D*). HT DNA transfection, but not lipofectamine alone, activated IFIT1-luc reporter expression, as expected, and this was also sensitive to wild type and VprR80A expression, but not expression of Vpr F34I/P35N (*Figure 3—figure supplement 1E,F*). Vpr Q65R had only a small inhibitory effect consistent with data in *Figure 3B*.

## Wild-type Vpr, but not sensing antagonism inactive Vpr mutants, colocalize with nuclear pores

Having identified Vpr mutants defective for antagonism of innate immune sensing, we sought further clues about Vpr mechanism by examining wild type and mutant Vpr location within cells. Vpr expressed in isolation is found in the nucleus and associated with nuclear pores (*Fouchier et al., 1998*; *Le Rouzic et al., 2002*). Concordantly, we found FLAG-Vpr in the nucleus, and colocalized with antibody staining the nuclear pore complex (NPC), when expressed by transient transfection in HeLa cells (*Figure 4A,B*). As previously reported for the single mutant F34I (*Jacquot et al., 2007*; *Vodicka et al., 1998*), we found that the double Vpr mutant F34I/P35N, as well as Vpr Q65R, were mislocalized, as compared to wild type and R80A Vpr. Thus, these mutants which fail to inactivate innate immune sensing, fail to localize to the nuclear membrane. Defective Vpr mutants F34I/P35N and Q65R appeared qualitatively different inside the nucleus, and nuclear rim staining was less well defined, suggesting that they have lost interactions with a protein(s) that normally influences their position within the cell. Fluorescence intensity measurements along transverse sections of nuclei in single confocal images showed two distinct peaks of nuclear pore staining representing each edge of the nucleus. These peaks overlapped with WT and Vpr R80A fluorescence but not with Vpr F34I/P35N or Vpr Q65R fluorescence, which was more diffuse and less well defined at the nuclear rim (*Figure 4C*). These data link Vpr nuclear membrane association with antagonism of innate immune sensing for the first time.

Vpr has been described to interact with cyclophilin A (CypA) and mutating Vpr residue P35 was reported to prevent this interaction (*Zander et al., 2003*). The nuclear pore complex has cyclophilin-like domains, which are structurally very similar to CypA, at the end of the Nup358 fibers that protrude into the cytoplasm (*Schaller et al., 2011*). To test whether Nup358 was required for Vpr association with the nuclear rim, we expressed FLAG-Vpr in Nup358-depleted HeLa cells (*Schaller et al., 2011*) and stained the Vpr FLAG tag (green) and NPC (red) (*Figure 4—figure supplement 1A,B*). Despite effective Nup358 depletion (*Figure 4—figure supplement 1C*), Vpr remained associated with the nuclear rim suggesting that Nup358 is not required for Vpr nuclear rim association (*Figure 4—figure supplement 1A,B,D*).

## Vpr inhibits IRF3 nuclear translocation

cGAMP is produced by activated cGAS and is recruited by STING, which then forms an active kinase complex in which TBK1 phosphorylates STING, TBK1 itself, and the transcription factor IRF3 (*Liu et al., 2015a*; *Zhang et al., 2019*). IRF3 phosphorylation promotes nuclear translocation and subsequent activation of gene expression including type 1 IFNs (*Chen et al., 2008*). As expected, transfection of THP-1 IFIT1-luc cells, with HT DNA, induced phosphorylation of STING, TBK1, and IRF3-S386 (*Figure 5A*). Measurement of IFIT1-luc expression, in the same samples, 3 hr after stimulation, indicated induction of IFIT1-luc by HT DNA, but not after prior Vpr expression using a lentiviral vector (*Figure 5B*). Strikingly, Vpr expression for 48 hr did not impact STING, TBK1, or IRF3 protein levels, or their phosphorylation status, 3 hr after DNA transfection, measuring IRF3 phosphorylation at S386 (*Figure 5A*). Empty vector expression had no detectable effect on protein levels or

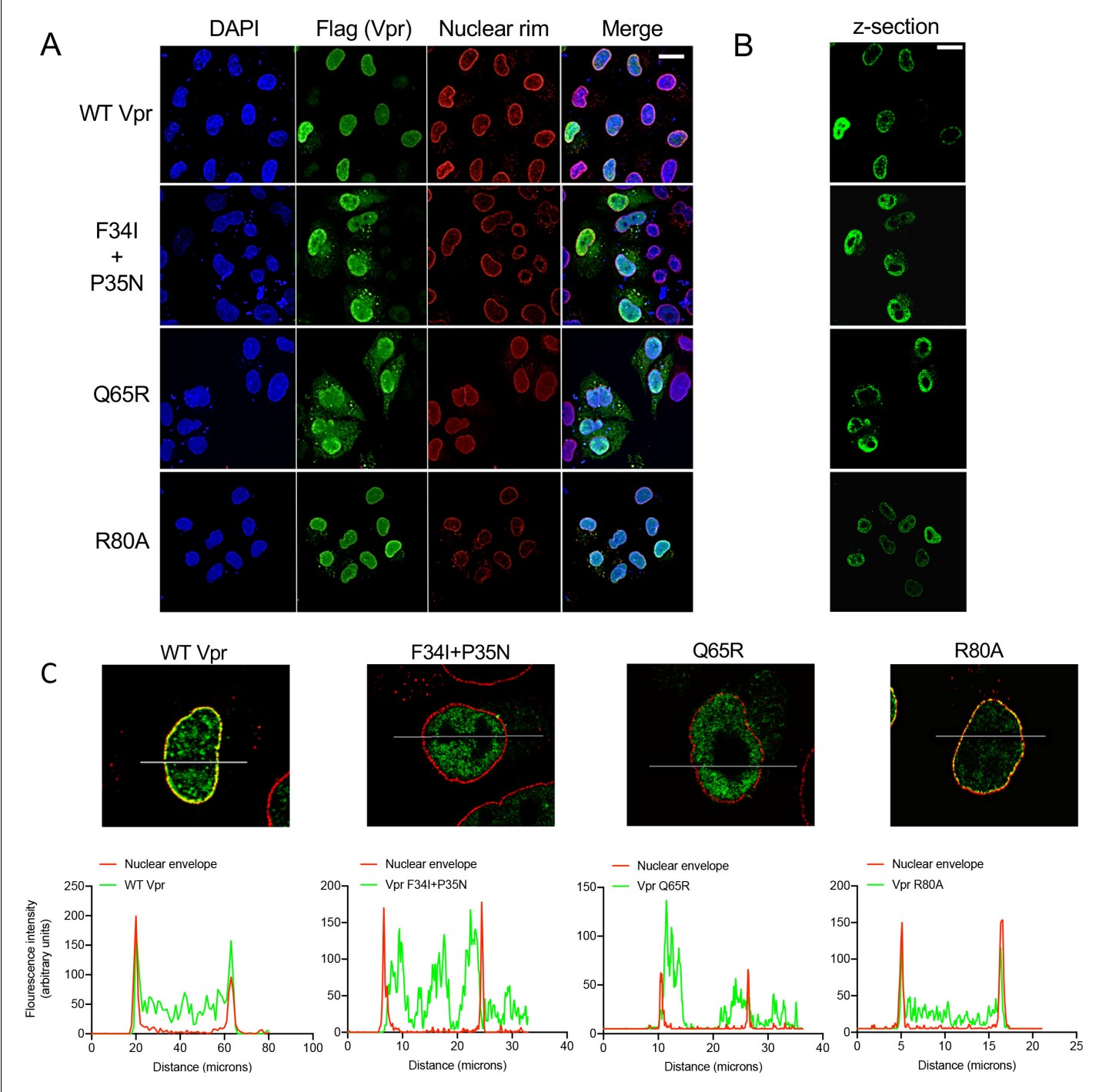

**Figure 4.** Wild-type Vpr, but not sensing antagonism inactive Vpr mutants, localize to nuclear pores. (A) Immunofluorescence confocal projections of HeLa cells transfected with Flag-tagged WT, or mutant, Vpr encoded by pcDNA3.1 plasmid (50 ng) and stained using antibodies detecting the Flag-tag (green) or nuclear pore complex (mab414) (red). 4',6-Diamidine-2'-phenylindole dihydrochloride (DAPI) stains nuclear DNA (Blue). (B) Selected confocal images (z-section) of cells in (A) showing effect of Vpr mutation on Vpr colocalization with mab414 nuclear pore staining. (C) Assessment of colocalization of Vpr with mab414 nuclear pore staining. Scale bars represent 10 μm.

The online version of this article includes the following figure supplement(s) for figure 4:

**Figure supplement 1.** Nup358 is not required for Vpr colocalization with mab414 nuclear pore staining.

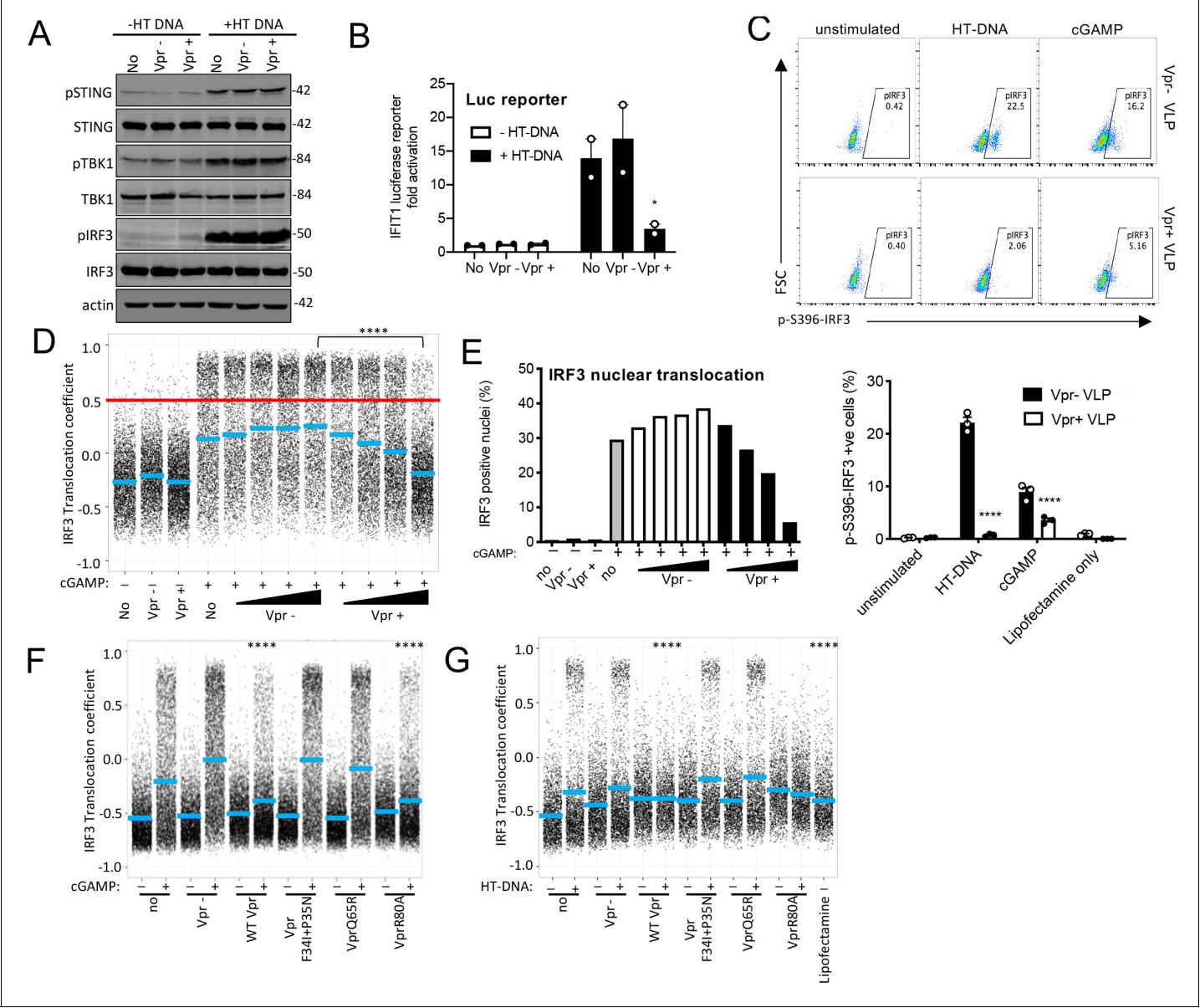

**Figure 5.** Vpr inhibits IRF3 nuclear translocation. (**A**) Immunoblot detecting Phospho-STING (Ser366), total STING, phospho-TBK1 (Ser172), total TBK1, phospho-IRF3 (Ser386), total IRF3, or actin as a loading control, from extracted THP-1 cells expressing Vpr from a lentiviral vector (MOI 1), expressing empty vector, or THP-1 left untransduced as a control and transfected with HT-DNA (5 μg/ml) or left untransfected as a control. Size markers are shown in kDa. (**B**) Mean fold induction of IFIT1-Luc in cells from A and B (**C**) Flow cytometry plot (forward scatter vs pIRF3-S396 fluorescence) of THP-1 cells infected with Vpr bearing virus-like particles (VLP) lacking genome (1 RT U/ml), or Vpr free VLP, stimulated with cGAMP (5 μg/ml) or HT-DNA transfection (5 μg/ml). Lower panel shows the flow cytometry data as a bar graph, plotting pIRF3-S396-positive cells. (**D**) Single-cell immunofluorescence measurement of IRF3 nuclear translocation in PMA differentiated THP-1 cells treated with cGAMP, or left untreated, and infected with HIV-1 GFP bearing Vpr, lacking Vpr or left untransduced. Cells were fixed and stained 3 hrs after infection/transfection. Red line shows the translocation coefficient threshold. Blue lines represent mean translocation coefficient. (**E**) Percentage of cells in D with IRF3 translocation coefficient greater than 0.5 (above red line). (**F**) Single-cell immunofluorescence measurement of IRF3 nuclear translocation in PMA-differentiated THP-1 cells stimulated with cGAMP (5 μg/ml), or left unstimulated, and infected with HIV-1 GFP lacking Vpr or bearing WT Vpr or Vpr mutants as shown (1 RT U/ml) or left uninfected. (**G**) Single cell immunofluorescence measurement of IRF3 nuclear translocation in PMA differentiated THP-1 cells transfected with HT-DNA (5 μg/ml), or left untransfected, and infected with HIV-1 GFP lacking Vpr, or bearing WT or mutant Vpr (1 RT U/ml) or left uninfected. Data in B is expressed as means ± SEM (n = 2). Data is analyzed using two-way ANOVA: * (p<0.05), ** (p<0.01), *** (p<0.001), **** (p<0.0001) compared to data from infection with HIV-1 lacking Vpr. Data are representative of three (**C–G**) or two (**A, B**) independent experiments.

The online version of this article includes the following figure supplement(s) for figure 5:

**Figure supplement 1.** Vpr inhibits IRF3 nuclear translocation.

*Figure 5 continued on next page*

*Figure 5 continued*

**Figure supplement 2.** Nuclear translocation of IRF3 after cGAMP stimulation in the presence or absence of Vpr.

phosphorylation (*Figure 5A*). Actin was detected as a loading control and Vpr/empty vector were used at a vector MOI of about 1 (*Figure 5—figure supplement 1A*). A second example of this experiment is presented in *Figure 5—figure supplement 1B–E*. IRF3 is phosphorylated at multiple sites during activation including at IRF3 S396. We therefore examined IRF3 S396 phosphorylation using a phospho-IRF3-S396 specific antibody and flow cytometry because this antibody did not work well by immunoblot. We found that in this case, Vpr delivery by VLP did reduce phosphorylation of IRF3-S396 after stimulation by either cGAMP or HT DNA in THP-1 cells (*Figure 5C*).

Given that Vpr is associated with the nuclear rim, and Vpr mutations that break antagonism of innate-sensing mislocalize Vpr, we hypothesized that rather than impacting levels of signaling proteins, Vpr may act at nuclear pores to influence nuclear transport of inflammatory transcription factors. This would be consistent with the broad innate immune antagonism that we have observed (*Figure 2*), and with previous reports of Vpr influencing nuclear transport, for example, of viral nucleic acids (*Heinzinger et al., 1994*; *Miyatake et al., 2016*; *Popov et al., 1998*), and inhibiting sensing of HIV-1 (*Trotard et al., 2016*). We therefore investigated the effect of Vpr on cGAMP-induced IRF3 nuclear translocation. THP-1 were differentiated with 50 ng/ml phorbol-12 myristate acetate (PMA) to attach them to glass for microscopy. In these experiments, VLP with or without Vpr are used to infect cells immediately after they are treated with innate immune stimulants. IRF3 translocation is measured three hours later by immunofluorescent labeling. VSV-G pseudotyped HIV-1 GFP-bearing Vpr reduced cGAMP-stimulated IRF3 nuclear translocation in a dose-dependent way, while HIV-1 lacking Vpr had no effect (*Figure 5D,E*, *Figure 5—figure supplement 2A*). These data are consistent with a previous report in which Vpr suppressed nuclear transport of IRF3-GFP on HIV-1 infection of HeLa cells in which DNA sensing had been reconstituted by expression of STING (*Trotard et al., 2016*). Importantly, in our experiments in THP-1, suppression of IRF3 nuclear translocation by Vpr was sensitive to Vpr mutation, with the same specificity as before (Compare *Figures 3*, *4* and *5F*, *Figure 5—figure supplement 1G–J*). HIV-1 GFP-bearing Vpr F34I/P35N, or Vpr Q65R, failed to efficiently suppress IRF3 nuclear localization after cGAMP stimulation (*Figure 5F*, S5G) or after transfection of differentiated THP-1 with HT DNA (*Figure 5G*, S5H). Conversely, HIV-1 GFP bearing wild-type Vpr, or Vpr R80A, effectively suppressed IRF3 nuclear localization after stimulation with cGAMP or HT DNA (*Figure 5F*, G S5G, H). Similar inhibition specificity by Vpr was also seen after activation of IRF3 nuclear translocation by transfection with the RNA mimic poly I:C (*Figure 5—figure supplement 1I,J*) or treatment with LPS (*Figure 5—figure supplement 1F*). Thus, suppression of IRF3 nuclear translocation correlates with the capacity of Vpr mutants to antagonize innate immune activation.

## Vpr inhibits NF-κB p65 nuclear translocation and NF-κB-sensitive plasmid expression

DNA sensing by cGAS is known to activate NF-κB as well as IRF3 (*Fang et al., 2017*). To test whether Vpr influenced NF-κB activation we repeated the experiment in *Figure 1C–F* but using THP-1 cells bearing an NF-κB-luciferase reporter (THP-1 NF-κB-luc) (*Figure 6A–C*). VSV-G pseudotyped HIV-1 GFP vector bearing Vpr minimally activated NF-κB-luc expression, whereas Vpr-negative HIV-1 GFP activated NF-κB-luc expression effectively (*Figure 6A*). Activation was dependent on viral genome because similar doses of HIV-1 VLP, made without genome, did not induce NF-κB-luc expression (*Figure 6A*). Viral doses were equalized by measurement of RT activity (SGPERT) (*Vermeire et al., 2016*). Vpr bearing, and Vpr negative, HIV-1 GFP were equally infectious and genome-free VLP were not infectious, as expected (*Figure 6B*). VSV-G pseudotyped HIV-1 GFP-bearing Vpr, but not virus lacking Vpr, suppressed cGAMP-mediated activation of the NF-κB-sensitive gene *IL6* (*Figure 6C*). We could not detect NF-κB nuclear localization in THP-1 after cGAMP treatment, perhaps due to timing, so we tested mutant Vpr specificity using Poly I:C to stimulate NF-κB p65 nuclear localization. Again, we transfected differentiated THP-1 cells, this time with Poly I:C and then immediately infected them with HIV-1 GFP bearing or lacking Vpr and fixed and stained for NF-κB p65 localization 3 hr later. We found Vpr inhibited NF-κB p65 nuclear localization with

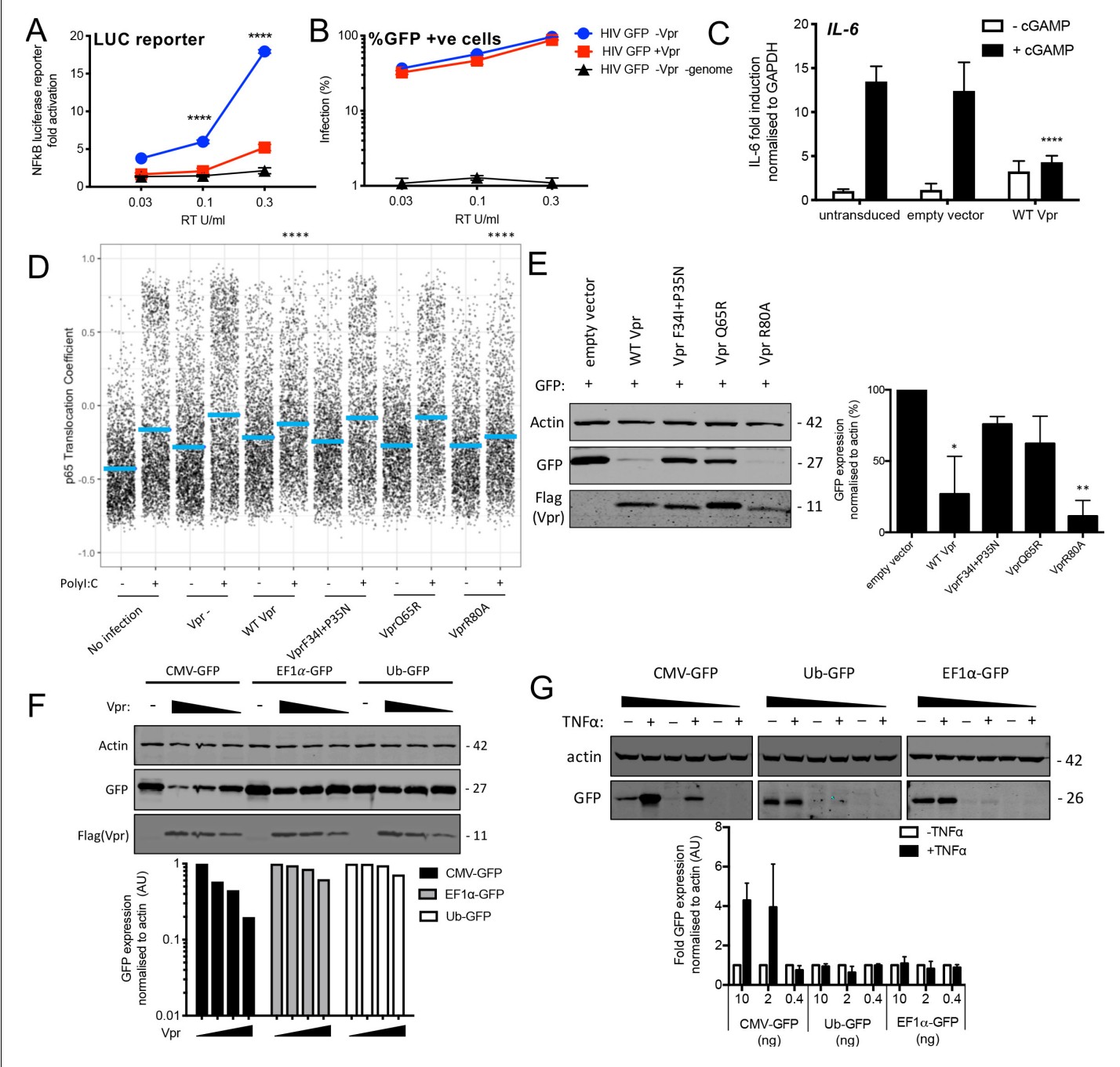

**Figure 6.** Vpr inhibits NF-κB p65 nuclear translocation and NF-κB-sensitive plasmid expression. (**A**) Fold induction of NF-κB-Luc after infection of THP-1 cells with HIV-GFP lacking Vpr, HIV-GFP bearing Vpr, or HIV-GFP lacking Vpr and genome, at the indicated doses. (**B**) Percentage of THP-1 cells in (**A**). (**C**) Fold induction of IL-6 after activation of STING by cGAMP (5 μg/ml) in cells expressing empty vector or Vpr encoding vector (MOI 1), or in untransduced THP-1 cells. (**D**) Single-cell immunofluorescence measurement of NF-κB (p65) nuclear translocation in PMA differentiated THP-1 cells transfected with Poly I:C (50 ng/ml), or left untreated, and infected with HIV-1 GFP lacking Vpr, HIV-1 GFP bearing Vpr (1 RT U/ml) or left uninfected. Cells were stained 3 hr after transfection and infection. (**E**) Immunoblot detecting Flag-Vpr, GFP, or actin as a loading control, from HEK293T cells transfected with 50 ng of empty vector, Flag-tagged WT Vpr vector, or Flag-tagged mutant Vpr vector, and CMV-GFP vector (50 ng). Size markers are shown in kDa. GFP expression from two independent immunoblots was quantified by densitometry and is shown in the lower panel. (**F**) Immunoblot detecting Flag-Vpr, GFP, or actin as a loading control, from HEK293T cells transfected with empty vector (200 ng) or Vpr vector (50 ng, 100 ng, 200 ng) and CMV-GFP, EF1α-GFP or Ub-GFP plasmids (50 ng). Size markers are shown in kDa. GFP expression quantified by densitometry is shown in the lower panel. (**G**) Immunoblot detecting GFP, or actin as a loading control, from HEK293T cells transfected with CMV-GFP, EF1α-GFP or Ub-GFP plasmids (10 ng, 2 ng, 0.4 ng) and stimulated with TNFα (200 ng/ml) or left unstimulated. Size markers are shown in kDa. GFP expression, from two independent

*Figure 6 continued on next page*

*Figure 6 continued*

immunoblots, quantified by densitometry, is shown in the lower panel. Data in (A, B, C) is expressed as mean ± SD (n = 3). Data in (E, F, G) is expressed as mean ± SD (n = 2). Two-way ANOVA: * (p<0.05), ** (p<0.01), *** (p<0.001), **** (p<0.0001) compared to empty vector or HIV GFP+Vpr.

The online version of this article includes the following figure supplement(s) for figure 6:

**Figure supplement 1.** Vpr inhibits NF-κB p65 nuclear translocation and NF-κB-sensitive plasmid expression.

similar sensitivity to mutation as for IRF3: VLP bearing wild-type Vpr or Vpr R80A inhibited NF-κB p65 nuclear localization but VLP bearing Vpr F34I/P35N or Vpr Q65R did not (*Figure 6D*, *Figure 6—figure supplement 1B*). Vpr also suppressed NF-κB p65 nuclear localization after treatment of THP-1 with LPS (*Figure 6—figure supplement 1C*).

Previous work has shown that Vpr inhibits the activity of the human CMV major immediate early promoter (MIEP) (*Liu et al., 2015a*). We hypothesized that this effect may be due to the dependence of this promoter on NF-κB (*DeMeritt et al., 2004*). As expected Flag-Vpr expression suppressed GFP expression from a co-transfected CMV MIEP – GFP construct (*Figure 6E*) as well as several other NF-κB-sensitive constructs expressing luciferase (*Figure 6—figure supplement 1A*). Importantly, Vpr mutants F34I/P35N, and Vpr Q65R suppressed GFP expression much less effectively than WT Vpr, or Vpr R80A, consistent with this effect being due to inhibition of NF-κB nuclear entry (*Figure 6E*, S6D, E). To probe this further, we used two constructs lacking NF-κB binding sites in which GFP is driven from the Ubiquitin C (Ub) promoter (*Matsuda and Cepko, 2004*) or from the elongation factor one alpha (EF1α) promoter (*Matsuda and Cepko, 2004*). Expression of GFP from these constructs was minimally affected by Vpr co-transfection, but GFP expression from the CMV MIEP was reduced as before (*Figure 6F*). Importantly, CMV MIEP-GFP expression was induced by activation of NF-κB with exogenous tumour necrosis factor alpha (TNFα), whereas Ub-GFP and EF1α-GFP were not, providing further evidence that Vpr inhibition correlated with promoter sensitivity to NF-κB (*Figure 6G*, *Figure 6—figure supplement 1F–G*). Thus, inhibition of NF-κB nuclear transport by Vpr likely explains the observation that Vpr suppresses expression from the CMV MIEP, but not promoters that are independent of NF-κB activity for expression. This is important because previous studies have used Vpr co-transfection with CMV MIEP driven promoters to address Vpr function (*Su et al., 2019*).

## HIV-1 Vpr interacts with karyopherins and inhibits NF-κB (p65) and IRF3 recruitment

WT Vpr suppresses nuclear entry of IRF3 and NF-κB, but Vpr DCAF1 binding mutant Q65R does not (*Figures 5* and *6*). This suggested that Vpr might degrade particular nuclear transport proteins to exert its effect. We therefore tested whether Vpr expression caused degradation of karyopherins KPNA1, KPNA2, KPNA3, KPNA4, KPNA5, KPNA6, or KPNB1. We infected cells with Vpr encoding HIV-1 vector, extracted total protein 48 hr after infection, and detected each protein using immunoblot (*Figure 7A*). However, we did not detect reduced levels of any of these karyopherins. It is possible that Vpr recruits karyopherins but does not degrade them. To test this, we sought interaction between Vpr and karyopherins KPNA1, KPNA2, and KPNA3 by co-immunoprecipitation. We found that immunoprecipitation of wild-type HA-Vpr co-precipitated Flag-KPNA1, as has been reported previously (*Miyatake et al., 2016*; *Nitahara-Kasahara et al., 2007*; *Vodicka et al., 1998*) and to a lesser degree Flag-KPNA2 and Flag-KPNA3, but not Flag-tagged GFP (*Figure 7B*). In a second experiment, we tested whether KPNA1-3 interacted with the inactive Vpr mutant F34I/P35N. WT Vpr interacted with KPNA1 as before, with less efficient interaction with KPNA2 and KPNA3 (*Figure 7C*). Importantly, KPNA1 interacted with the Vpr F34I/P35N only very weakly, and much less than WT Vpr, consistent with the mutant's reduced activity in antagonizing innate immune sensing (*Figure 7C*). Given that Vpr expression did not cause KPNA1 degradation, we sought evidence for Vpr disturbing interactions between KPNA1 and IRF3 or NF-κB p65. HA-IRF3 immunoprecipitated with Flag-KPNA1 as expected and this interaction was reduced by expression of WT Vpr, but not inactive mutant Vpr F34I/P35N (*Figure 7D*). A competing immunoprecipitation experiment with KPNA1 and NF-κB p65 gave similar results. Immunoprecipitation of Flag-KPNA1 co-precipitated NF-κB p65 and this was reduced by co-expression of WT Vpr, but not Vpr F34I/P35N (*Figure 7E*). Thus, for the first time, we explain the interaction of Vpr with karyopherins, by demonstrating that it

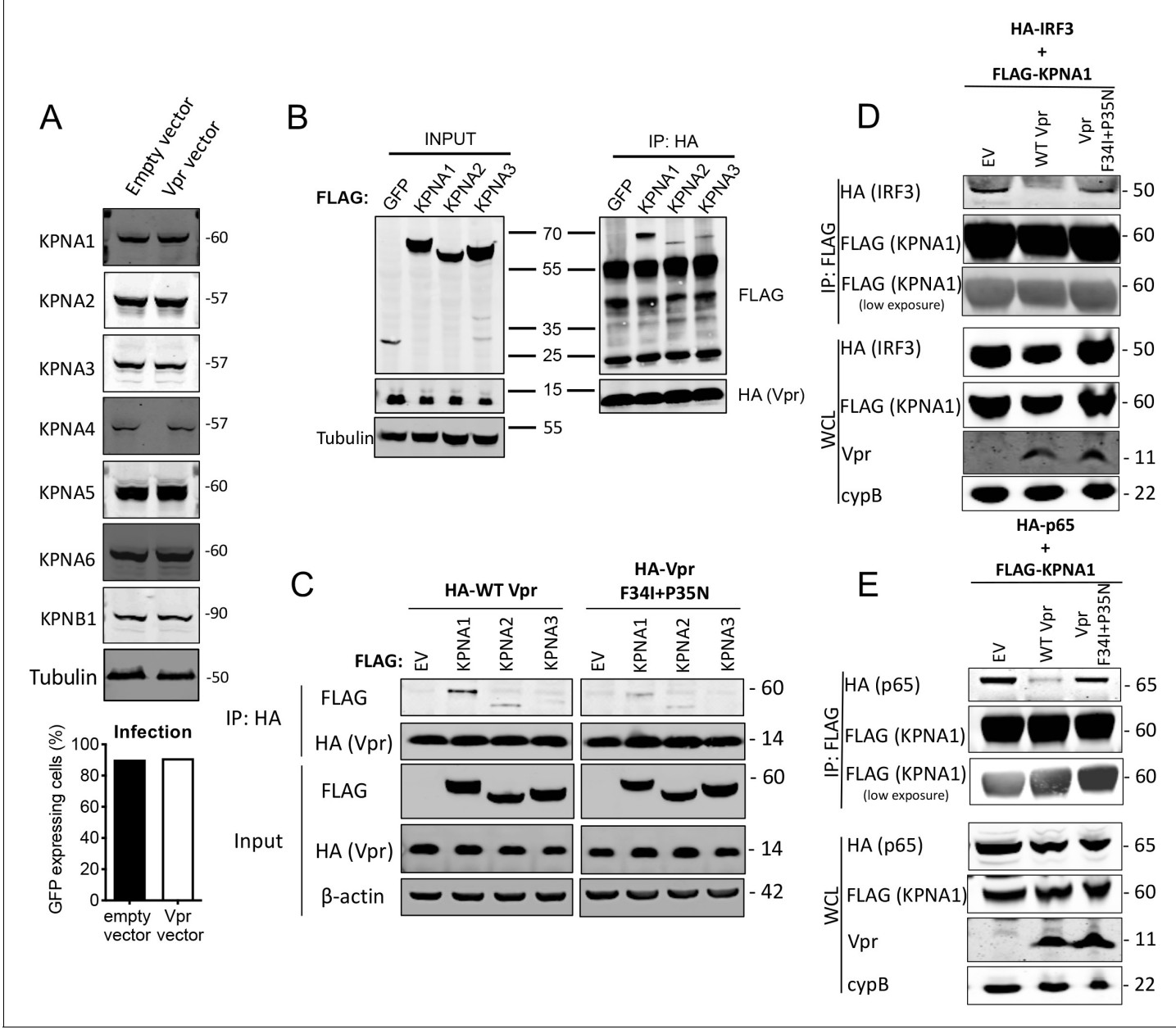

**Figure 7.** HIV-1 Vpr interacts with karyopherins and inhibits IRF3/NF-κB(p65) recruitment to KPNA1. (**A**) Immunoblot detecting KPNA1-6 or KPNB1 from extracted HEK293T cells infected with empty vector, or Vpr encoding vector at a dose of 0.05 RT U/ml (MOI = 2). Size markers are shown in kDa. Percentage infection by HIV-1 GFP bearing Vpr encoding or empty vector is shown on the right. (**B**) Co-immunoprecipitation of Flag-KPNA1-3 and HA-Vpr. Input shows immunoblot detecting extracted HEK293T whole cell lysates expressing flag-KPNA1-3, flag-GFP and HA-Vpr before immunoprecipitation. Co-immunoprecipitation precipitates Vpr with HA-beads and detects Flag-KPNA1-3. (**C**) Co-immunoprecipitation of Flag-KPNA1-3 and WT HA-Vpr or HA-Vpr F34I+P35N. Input shows immunoblots detecting HA-Vpr or Flag-KPNA1-3 in extracted HEK293T whole cell lysates (WCL) before immunoprecipitation. β-Actin is detected as a loading control. Co-immunoprecipitation precipitates Vpr with HA-beads and detects Flag-KPNA1-3. (**D**) Co-immunoprecipitation of HA-IRF3 and Flag-KPNA1 in the presence and absence of WT Vpr or Vpr F34I+P35N to detect competition between Vpr and IRF3 for KPNA1. Input shows immunoblots detecting HA-IRF3 or Flag-KPNA1 or Vpr in extracted HEK293T whole cell lysates (WCL) before immunoprecipitation. CypB is detected as a loading control. Co-immunoprecipitation precipitates KPNA1 with Flag-beads and detects HA-IRF3 in the presence and absence of WT Vpr or inactive Vpr F34I+P35N. (**E**) Co-immunoprecipitation of HA-NF-κB p65 and Flag-KPNA1 in the presence and absence of WT Vpr or Vpr F34I+P35N to detect competition between Vpr and p65 for KPNA1. Input shows immunoblots detecting HA-p65 or Flag-KPNA1 or Vpr in extracted HEK293T whole cell lysates (WCL) before immunoprecipitation. CypB is detected as a loading control. Co-immunoprecipitation precipitates KPNA1 with Flag-beads and detects HA-p65 in the presence and absence of WT Vpr or Vpr F34I+P35N. The online version of this article includes the following figure supplement(s) for figure 7:

**Figure supplement 1.** A unifying model of Vpr function.

prevents them from efficiently recruiting and transporting transcription factors IRF3 and NF-κB into the nucleus after innate immune activation. This finding provides a mechanistic basis for the broad innate immune antagonism activity of Vpr and links manipulation of nuclear transport with antagonism of innate immunity rather than with infection itself.

## Discussion

Despite many studies investigating Vpr function, a clear mechanism for how HIV-1 Vpr promotes replication has not been forthcoming, partly because Vpr replication phenotypes have not been clearly mechanistically linked to manipulation of specific target proteins. Early work connected nuclear membrane association of Vpr with replication in macrophages, but not T cells (*Connor et al., 1995*; *Dedera et al., 1989*; *Fouchier et al., 1998*; *Hattori et al., 1990*; *Mashiba et al., 2015*; *Vodicka et al., 1998*). Early work also separated the effect of Vpr on cell cycle from its association with the nuclear envelope using Vpr mutants, particularly Vpr F34I, which, as confirmed herein, suppressed cell cycle, but did not recruit to the nuclear membrane (*Jacquot et al., 2007*; *Vodicka et al., 1998*). Vpr mutants that did not localize to the nuclear membrane, did not promote macrophage replication, leading the authors to reasonably conclude that Vpr contributed to nuclear transport of the virus itself. This observation was consistent with the notion that Vpr-mediated support of nuclear entry is expected to be more important in non-dividing cells (macrophages), than rapidly dividing cells (activated T cells). Vpr is also not typically required for infection of cell lines, even if they are not dividing (*Yamashita and Emerman, 2005*).

In complementary studies, Vpr has been associated with antagonism of innate immune sensing in macrophages (*Harman et al., 2015*), T cells (*Vermeire et al., 2016*), as well as in HeLa cells reconstituted for DNA sensing by STING expression (*Trotard et al., 2016*). Here we propose a model that unifies Vpr's role in manipulating nuclear entry with its antagonism of innate immune signaling. We propose that Vpr interaction with karyopherin KPNA1 (*Figure 7*; *Miyatake et al., 2016*; *Nitahara-Kasahara et al., 2007*; *Vodicka et al., 1998*) inhibits nuclear transport of activated IRF3 and NF-KB (*Figures 5–7*) and subsequent gene expression changes downstream of innate immune sensing (*Figures 1–3*). Thus, HIV-1 Vpr antagonizes the consequences of innate immune activation by HIV-derived, and non-HIV derived PAMPs alike. This explains its importance for maximal replication in macrophages, because activated T cells, and most cell lines, respond to innate immune agonists poorly, and particularly to DNA-based PAMPs (*Figure 1*; *Cingöz and Goff, 2019*; *de Queiroz et al., 2019*; *Heiber and Barber, 2012*; *Xia et al., 2016a*; *Xia et al., 2016b*).

We propose that previous demonstrations of Vpr-dependent HIV-1 replication in macrophages, that depended on association of Vpr with the NPC, or with nuclear transport factors, are explained by Vpr inhibition of innate immune sensing and subsequent antiviral responses (*Jacquot et al., 2007*; *Vodicka et al., 1998*). Indeed, we now know that induction of an innate response by HIV-1 lacking Vpr is expected to suppress viral nuclear entry because IFN induction of MxB in macrophages causes inhibition of HIV-1 nuclear entry (*Goujon et al., 2013*; *Kane et al., 2013*). Thus, we propose that Vpr does not directly promote HIV-1 nuclear entry. Rather it prevents inhibition of nuclear entry downstream of innate immune activation. We hypothesize that Vpr provides an *in vivo* replication advantage because activation of IRF3 and NF-KB induces expression of inflammatory cytokines, including type 1 IFNs, and subsequently restriction factors for which HIV-1 does not encode antagonists. For example, in addition to MxB, IFN induces IFITM1-3 (*Foster et al., 2016*), and TRIM5α (*Jimenez-Guardeño et al., 2019*) all of which can inhibit HIV-1. Concordantly, accidental infection of a lab worker with a Vpr-defective HIV-1 isolate resulted in delayed seroconversion, suppressed viremia, and normal T-cell counts without need for antiviral treatment (*Ali et al., 2018*).

In most of the experiments herein, and in previous studies of Vpr function in cell lines (*Yamashita and Emerman, 2005*), Vpr did not impact infection of single round VSV-G pseudotyped HIV-1 vectors encoding GFP. We propose that this is because if antiviral inflammatory responses, for example IFN, are triggered at around the time of infection, either by exogenous signals, or by HIV-1 itself, then the activated antiviral effectors are too slow to inhibit that infection, that is, the expression of GFP from an integrated provirus. Thus, a requirement for Vpr is only revealed by spreading infection assays in innate competent cells such as macrophages, which can suppress replication of subsequent rounds of infection.

We and others, have argued that the wild-type infectious HIV-1 genome is not efficiently sensed by nucleic acid sensors, or degraded by cellular nucleases, because the capsid protects and sequesters genome, while regulating the process of reverse transcription, during transport across a hostile cytoplasmic environment, prior to uncoating at the NPC, or in the nucleus of infected cells (*Bejarano et al., 2019*; *Burdick et al., 2017*; *Francis et al., 2016*; *Jacques et al., 2016*; *Rasaiyaah et al., 2013*; *Schaller et al., 2011*; *Sumner et al., 2020*; *Towers and Noursadeghi, 2014*; *Yan et al., 2010*; *Zila et al., 2019*). Indeed, we find that Vpr can promote HIV-1 replication, even if the innate immune stimulation does not originate from an HIV-1-derived PAMP, here exemplified by replication assays in cGAMP-treated primary human macrophages (*Figure 1*). We also found that Vpr antagonized the effects of exposure to LPS, RNA and DNA ligands, as well as other viral infections, exemplified here by Sendai virus infection, which potently activates RNA sensing and IFN production in human macrophages (*Matikainen et al., 2000*; *Figure 2*). In this way, Vpr can suppress activation signals connected indirectly to infection. A series of recent studies have demonstrated that infected cells produce a diverse range of endogenous RNA- and DNA-derived PAMPs. Examples include retroelement induction by influenza infection (*Schmidt et al., 2019*), RNA pseudogene expression after herpes simplex virus infection (*Chiang et al., 2018*) and RIGI ligands after Kaposi's sarcoma herpes virus infection (*Zhao et al., 2018*). These studies suggest that viruses must be able to manage innate activation from non-viral PAMPs even when their own PAMPs are sequestered. HIV-1 infection has also been described to induce retroelement expression (*Jones et al., 2013*) consistent with a requirement for Vpr to suppress innate immune activation downstream of endogenous PAMPs. Furthermore, HIV seroconversion has been associated with a cytokine storm (*Stacey et al., 2009*) the antiviral effect of which may be mitigated by particle associated Vpr. Thus, HIV-1 may utilize Vpr to replicate in an innate immune activated environment, even when its own PAMPs are effectively sequestered. A link between escape from innate sensing and successful transmission is suggested by several lines of evidence. These include a generally low HIV transmission frequency (*Shaw and Hunter, 2012*), the observation that HIV transmitted founder clones are particularly resistant to IFN (*Iyer et al., 2017*), and encode distinct Vpr amino acid signatures, as compared to chronic viruses (*Rossenkhan et al., 2016*), as well as the HIV transmission-associated cytokine storm itself (*Stacey et al., 2009*). Concordantly, Vpu, Nef and Vif, and Vpr, antagonize innate immunity to enhance viral replication, reviewed in *Sumner et al., 2020*.

Vpr has been suggested to cause IRF3 degradation (*Okumura et al., 2008*), but we did not detect IRF3 degradation in THP-1 cells under conditions when gene expression and IRF3 nuclear transport were strongly suppressed (*Figure 5*). Furthermore, in addition to suppressing IRF3 nuclear transport, we found that Vpr reduced IRF3 phosphorylation at S396 but not at S386 (*Figure 5*). Previous studies have suggested that phosphorylation of IRF3 at S386 is necessary and sufficient for IRF3 activation (*Lin et al., 1999*; *Mori et al., 2004*; *Schirrmacher, 2015*; *Servant et al., 2003*; *Suhara et al., 2000*; *Yoneyama et al., 1998*). Thus, our data are consistent with a more complex picture of IRF3 activation by phosphorylation. It is possible that phosphorylation at S396 occurs in a karyopherin or NPC-dependent way that is occluded by Vpr recruitment to karyopherin. Phosphorylation of IRF3 at S396 has been associated with enhanced association and multimerization with transcriptional coactivator CREB binding protein (CBP/p300) suggesting a later role than phosphorylation at S386 (*Chen et al., 2008*). It is possible that the lack of S396 IRF3 phosphorylation is a consequence of IRF3 dephosphorylation at S396 as nuclear entry is prevented.

Inhibition of IRF3 phosphorylation is also consistent with reported inhibition of TBK1 by Vpr, although this study detected inhibition of TBK1 phosphorylation, whereas we did not (*Harman et al., 2015*). In that study, Vpr promoted infection in macrophages and dendritic cells, despite HIV induced formation of innate immune signaling complexes containing TBK1, IRF3, and TRAF3, visualized by immunofluorescence staining. Thus, TBK1 inhibition by Vpr may occur in addition to Vpr activity on nuclear transport, because TBK1 is seen in the cytoplasm, not at the nuclear envelope, in these HIV-infected cells (*Harman et al., 2015*). IRF3 degradation was not detected in this study and nor was HIV-1 induced IRF3 phosphorylation, although the impact of infection on IRF3 by wild-type HIV-1 and HIV-1 deleted for Vpr were not compared.

The regulation of the nuclear import of NF-KB and IRF3 by multiple karyopherins is expected to be complex (*Fagerlund et al., 2005*; *Fagerlund et al., 2008*; *Kumar et al., 2000*; *Liang et al., 2013*). Targeting karyopherins is a typical viral strategy for manipulation of cellular responses but the different ways viruses perform this function hints at the complexity required to inhibit innate

responses whilst avoiding shutting down viral transcription. We propose that the different mechanisms of NF-κB/IRF3 manipulation by different viruses reflect their reliance on transcriptional activation while simultaneously depending on inhibition of the same transcription factors activated by defensive processes. We hypothesize that each virus has specifically adapted to facilitate replication while dampening activation of inhibitory effectors. Failure to degrade karyopherin proteins suggests that some KPNA1 nuclear import function may be left intact by HIV to facilitate a more subtle manipulation of host cell biology (*Figure 7*). A similar model of inhibition of KPNA target binding to manipulate nuclear import has been suggested by a crystal structure of Ebola Virus VP24 protein in complex with KPNA5. This study proposed that VP24 targets a KPNA5 NLS binding site to specifically inhibit nuclear import of phosphorylated STAT1 (*Xu et al., 2014*).

Cell type clearly also plays a role in Vpr function. For example, in myeloid cells (*Kogan et al., 2013*; *Miller et al., 2017*), and T cells (*Ayyavoo et al., 1997*), Vpr has been reported to inhibit NF-KB. Other studies in T cells suggest NF-KB activation by Vpr to drive viral transcription (*Liu et al., 2014*; *Vermeire et al., 2016*). In a more recent study, Hotter and colleagues showed that expression of diverse primate immunodeficiency virus Vprs in HEK293T cells could activate or inhibit NF-KB activity depending on the assay (*Hotter et al., 2017*). For example, Vpr expression in HEK293T cells activated baseline, and TNFα stimulated, expression of a transfected NF-KB-sensitive reporter, but inhibited activation of reporter by transfected IKKβ. The authors proposed that Vpr-mediated inhibition of NF-KB was relevant because Vpr inhibited an IFNβ reporter activated by Sendai Virus infection, consistent with results presented herein. We propose that cell type, and the stage of the viral life cycle, influence the effect of Vpr on transcription factor activation. One possibility is that incoming particle associated Vpr is active against NF-KB, to mitigate innate sensing, but Vpr expressed from the provirus in an infected cell is bound by Gag, which sequesters Vpr, reducing further inhibition of the activated NF-KB that is required for on-going viral transcription (*Belzile et al., 2010*).

Our data also explain previous reports of the suppression of expression from co-transfected CMV MIEP-driven plasmids by Vpr (*Liu et al., 2015b*). Vpr inhibition of NF-KB transport into the nucleus to activate the MIEP likely explains these data, but another possibility is that transcription factor bound to cytoplasmic plasmid DNA has a role in importing plasmid into the nucleus, and it is plasmid transport that is inhibited (*Mesika et al., 2001*). Vpr insensitivity of NF-KB-independent ubiquitin and EF1α promoters (*Figure 6*) is consistent with this model, summarized in *Figure 7—figure supplement 1A*. This is important because inhibition of transfected plasmid driven protein expression may explain the effect of cotransfected SIV Vpr on STING and cGAS signaling reported recently (*Su et al., 2019*). Note that STING expression was not affected by Vpr co-expression but STING was expressed from the Vpr and NF-KB-insensitive EF1α promoter (*Figure 6*), whereas cGAS, which was not measured by western blot, was expressed from a Vpr and NF-KB-sensitive (*Figure 6*) CMV-driven plasmid VR1012 (*Hartikka et al., 1996*). Some experiments in *Hotter et al., 2017* may also have been influenced by this phenomenon.

Importantly, our data are consistent with reports that manipulation of cell cycle by Vpr is independent of interaction with karyopherin proteins. The Vpr R80A mutant, which does not arrest cell cycle, or manipulate SLX4 complex (*Gaynor and Chen, 2001*; *Laguette et al., 2014*) was functional in inhibition of innate sensing (*Figures 3*, *5* and *6*). Thus we assume that SLX4 interaction does not play a role in the innate immune antagonism shown herein. Mapping the residues of Vpr that are important for innate immune inhibition onto structures resolved by NMR and X-ray crystallography reveals a potentially distinct interface from that targeting UNG2 because residues Vpr 34/35 are distant from the UNG2 binding site (*Figure 3—figure supplement 1B,C*). Further, UNG2 has not been associated with innate immune sensing. Given that Vpr has been shown to bind FxFG motif in p6 of Gag during virion incorporation (*Zhu et al., 2004*), and FG motifs at the NPC (*Fouchier et al., 1998*) it is possible that interaction of Vpr with nuclear pore proteins via the FG motifs contribute to Vpr-mediated inhibition of IRF3 and NF-KB nuclear import.

*In vitro*, primary myeloid cells behave according to the stimuli they have received. Thus, inconsistent results between studies, for example the requirement here for cGAMP, but not in other studies, to cause Vpr-dependent replication in macrophages (*Figure 1*), could be explained by differences in myeloid cell stimulation due to differences in cell purification and differentiation methods or reagents used. Methods of virus preparation, here viruses were purified by centrifugation through sucrose, may also be a source of target cell activation and experimental variation. We hypothesize that cGAMP induced Vpr dependence in MDM (*Figure 1*) because cells were not activated prior to

cGAMP addition, whereas in other studies basal activation produced Vpr-dependent replication. Replication in activated primary CD4+ T cells was, in our hands, independent of Vpr in the presence and absence of cGAMP, which was inhibitory, suggesting that Vpr cannot overcome signaling downstream of cGAMP in these cells. This implies that activated T-cells respond differently to cGAMP than macrophages, consistent with observations that in T cell/macrophage mixed cultures, the negative effects of cGAMP on HIV-1 replication were principally mediated via macrophages (*Xu et al., 2016*). Vpr-sensitive, cGAS-dependent, IFN production from T cells has been reported suggesting that in the right circumstances, T cells can sense HIV-1 DNA, via cGAS, in T cells (*Vermeire et al., 2016*). Importantly, this study used integration inhibition to demonstrate provirus-dependent detection of HIV-1 suggesting that incoming HIV-1 DNA is not the cGAS target in this study. The nature of the PAMP in these experiments remains unclear. Certainly, further work is required to understand the different requirements for Vpr function in T cells and macrophages.

Sensing of HIV-1 is clearly viral dose, and therefore PAMP dose, dependent. For example, Cingoz et al reported failure of VSV-G pseudotyped HIV-1 (ΔEnv, ΔNef, ΔVpr) to activate sensing in a variety of cell lines (*Cingöz and Goff, 2019*). However, other studies have demonstrated sensing of wild-type HIV-1 DNA by cGAS (*Gao et al., 2013*; *Lahaye et al., 2013*), and here we observed cGAS-dependent, Vpr-sensitive, induction of CXCL10 or NF-KB reporter by high dose (MOI 1–3) VSV-G pseudotyped single round HIV-1 GFP vector in THP-1 cells (*Figures 1* and *6*). The effect of dose is illustrated in *Figure 1* in which MOI (0.1–0.3) had little effect on CXCL10 expression in THP-1 cells. However, higher doses activated CXCL10 expression, unless the virions carried Vpr, in which case CXCL10 induction was suppressed. Cingoz used luciferase to measure infection and therefore MOIs are obscure making dose comparison difficult. Note that herein, MOI calculated by GFP expression are included in supplementary data for most experiments. Given that infection typically depends on exposing cells to more than one viral particle, requiring tens of particles in even the most conservative estimates, it is likely that Vpr delivered by particles, that do not eventually form a provirus, contributes to suppression of sensing. Certainly, a lower MOI is required for Vpr activity when the stimulation comes from the Vpr bearing viral particles themselves (MOI three required, *Figure 1C*), compared to from external stimulus (MOI 20 required, *Figure 1B*). It is hard to know what MOI are really relevant to replication *in vivo*, but it is important to note that in our experiments, high MOI above one are required for innate immune activation and Vpr-dependent antagonism. This suggests that low MOI infection depends on sensor evasion by viral PAMP sequestration within intact capsids (*Jacques et al., 2016*) but higher MOI infections can rely on particle associated Vpr to suppress the activation of any exposed viral PAMPs and any endogenous PAMPs that are induced.

In summary, our findings connect Vpr manipulation of nuclear transport with inhibition of innate immune sensing, rather than viral nuclear import. They highlight the crucial role of particle associated Vpr in inhibiting innate immune activation during the early stages of the viral life cycle and unify a series of studies explaining previously apparently unconnected observations. Given the complexity of NF-kB activation, and the different ways each virus manipulates defensive transcriptional responses, we propose that the further study of viral inhibition of PAMP-driven inflammatory responses will lead to a better understanding of the biology of the transcription factors involved and highlight novel, tractable targets for therapeutic antiinflammatory development.

# Materials and methods

**Key resources table**

| Reagent type (species) or resource | Designation | Source or reference | Identifiers | Additional information |
|---|---|---|---|---|
| Antibody | anti-FXFG repeats (mouse monoclonal) | Abcam | Cat# ab24609 | IF (1:1000) |
| Antibody | Anti-FLAG tag (mouse monoclonal) | Sigma | Cat# F3165 | IF (1:1000) |
| Antibody | Anti-IRF3 (rabbit polyclonal) | Santa Cruz biotechnology | Cat# sc-9082 | IF (1:400) |

*Continued on next page*

*Continued*

| Reagent type (species) or resource | Designation | Source or reference | Identifiers | Additional information |
| --- | --- | --- | --- | --- |
| Antibody | Anti-rabbit alexa fluor 488 IgG (goat polyclonal) | Invitrogen | Cat# A-11008 | IF (1:500) |
| Antibody | Anti-mouse Alexa Fluor 546 IgG (goat polyclonal) | Invitrogen | Cat# A-11030 | IF (1:500) |
| Antibody | Anti-VSV-G (rabbit polyclonal) | Sigma | Cat# V4888 | WB (1:20,000) |
| Antibody | Anti-HIV-1 p24 (mouse monoclonal) | NIH AIDS reagent program | Cat# 3537 | WB (1:1000) |
| Antibody | Anti-STING (Rabbit monoclonal) | Cell Signaling | Cat# 13647 | WB (1:1000) |
| Antibody | Anti-phospho STING (Rabbit monoclonal) | Cell Signaling | Cat# 19781 | WB (1:1000) |
| Antibody | Anti-TBK1 (Rabbit monoclonal) | Cell Signaling | Cat# 3504S | WB (1:1000) |
| Antibody | Anti-phospho TBK1 (Rabbit monoclonal) | Cell Signaling | Cat# 5483 | WB (1:1000) |
| Antibody | Anti-IRF3 (Rabbit monoclonal) | Cell Signaling | Cat# 4302 | WB (1:1000) |
| Antibody | Anti-phospho -IRF3 S386 (Rabbit monoclonal) | Abcam | Cat# ab76493 | WB (1:1000) |
| Antibody | Anti-phospho- IRF3 S396 (Rabbit monoclonal) | Cell signaling | Cat# D6O1M | Flow Cytometry (1:50) |
| Antibody | Anti-actin (mouse polyclonal) | Abcam | Cat# ab8227 | WB (1:20,000) |
| Antibody | Anti-cGAS (rabbit monoclonal) | Cell Signaling Technology | Cat# 15102 | WB (1:1000) |
| Antibody | Anti-MAVS (mouse polyclonal) | Cell Signaling Technology | Cat# 3993 | WB (1:1000) |
| Antibody | Anti-DCAF1(rabbit polyclonal) | Bethyl | Cat# A301-887A | WB (1:1000) |
| Antibody | Anti-Nup358 (rabbit polyclonal) | Abcam | Cat# ab64276 | WB (1:1000) |
| Antibody | Anti-FLAG (mouse monoclonal) | Sigma | Cat# F3165 | WB (1:1000) |
| Antibody | Anti-GFP (rabbit polyclonal) | Abcam | Cat# ab6556 | WB (1:20,000) |
| Antibody | Anti-KPNA1 (rabbit polyclonal) | ABclonal | Cat# A1742 | WB (1:1000) |
| Antibody | Anti-KPNA2 (rabbit polyclonal) | ABclonal | Cat# A1623 | WB (1:1000) |
| Antibody | Anti-KPNA3 (rabbit polyclonal) | ABclonal | Cat# A8347 | WB (1:1000) |
| Antibody | Anti-KPNA4 (rabbit polyclonal) | ABclonal | Cat# A2026 | WB (1:1000) |
| Antibody | Anti-KPNA5 (rabbit polyclonal) | ABclonal | Cat# A7331 | WB (1:1000) |
| Antibody | Anti-KPNA6 (rabbit polyclonal) | ABclonal | Cat# A7363 | WB (1:1000) |
| Antibody | Anti-KPNB1 (rabbit polyclonal) | ABclonal | Cat# A8610 | WB (1:1000) |
| Antibody | Anti-CypB (rabbit polyclonal) | Abcam | Cat# ab16045 | WB (1:5000) |
| Antibody | Anti-HA (rabbit polyclonal) | Sigma | Cat# H6908 | WB (1:1000) |
| Antibody | Anti-Vpr (rabbit polyclonal) | NIH AIDS reagents programme | Cat# 11836 | WB (1:1000) |
| Antibody | Anti-mouse IgG IRdye 800CW (goat poly clonal) | LI-COR Biosciences | Cat# 926–32210 | WB (1:10,000) |
| Antibody | Anti-rabbit IgG IRdye 800CW (goat poly clonal) | LI-COR Biosciences | Cat# 926–32211 | WB (1:10,000) |
| Other | Herring testes DNA | Sigma | Cat# D6898 | Amount used stated in text |
| Other | cGAMP | Invivogen | Cat code (tlrl-nacga23-1) | Amount used stated in text |
| Other | Poly I:C | Invivogen | Cat code (tlrl-pic) | Amount used stated in text |
| Other | Lipopolysaccaride | Invivogen | Cat code (tlrl-smlps) | Amount used stated in text |

## Cells and reagents

HEK293T cells were maintained in DMEM (Gibco) supplemented with 10% fetal calf serum (FCS, Labtech) and 100 U/ml penicillin and 100 µg/ml streptomycin (Pen/Strep; Gibco). THP-1 cells were maintained in RPMI (Gibco) supplemented with 10% FCS and Pen/Strep. THP-1-IFIT-1 luciferase reporter cells express Gaussia luciferase under the control of the endogenous IFIT1 promoter have been described (*Mankan et al., 2014*). THP-1 CRISPR control, *cGAS-/-* and *MAVS -/-* knock out cells have been described (*Mankan et al., 2014*). Nup358 depleted HeLa cells have been described (*Schaller et al., 2011*). Lipopolysaccharide, poly I:C and TNFα were obtained from PeproTech. Sendai virus was obtained from Charles River Laboratories. Herring-testis DNA was obtained from Sigma. cGAMP was obtained from Invivogen. NF-KB Lucia THP-1 reporter cells were obtained from Invivogen. All cell lines were tested negative for mycoplasma.

## Cloning and plasmids

The Vpr gene from HIV-1 founder clone SUMA (*Fischer et al., 2010*) was codon optimized and synthesized by GeneArt. To generate the HIV-1 vector encoding Vpr (pCSVIG), the codon optimized SUMA Vpr gene was cloned into pSIN-BX-IRES-Em between BamHI and XhoI sites under the control of the SFFV LTR promoter. pSIN-BX-IRES-Em was obtained from Dr Yasuhiro Takeuchi. EF1α-GFP and UB-GFP were obtained from Addgene (*Matsuda and Cepko, 2004*). The CMV-GFP construct was pEGFPC1 (Clontech). HIV-1 bearing a Ba-L envelope gene has been described (*Rasaiyaah et al., 2013*). Flag- KPNA1-3 plasmids were obtained from Prof. Geoffrey Smith. HIV-1ΔVpr was a gift from Richard Sloan and encoded an 17 nucleotide insertion (Vpr 64–81) that destroys the Vpr coding sequence.

## Production of virus in HEK293T cells

Replication competent HIV-1 and VSV-G pseudotyped HIV-1 GFP vectors were produced by transfection of HEK293T cells in T150 flasks using Fugene six transfection reagent (Promega) according to the manufacturer's instructions. Briefly, just-subconfluent T150 flasks were transfected with 8.75 µg of HIV-1 YU2 or HIV-1 YU2 lacking Vpr (HIV-1 YU2 ΔVpr) and 30 µl Fugene 6 in 500 µl Optimem (Thermofisher Scientific). To make VSV-G pseudotyped HIV-1 GFP, each T150 flask was transfected with 2.5 µg of vesicular stomatitis virus-G glycoprotein encoding plasmid (pMDG) (Genscript), 2.5 µg of packaging plasmid, p8.91 (encoding Gag-Pol, Tat and Rev) or p8.2 (encoding Gag-Pol, Tat and Rev and Vif, Vpr, Vpu and Nef) (*Zufferey et al., 1997*), and 3.75 µg of GFP encoding genome plasmid (pCSGW) using 30 µl Fugene 6 in 500 µl optimum. To make Vpr encoding HIV-1 GFP, 3.75 µg pCSVIG was transfected with 2.5 µg of pMDG and 2.5 µg of p8.91. To make HIV-1 GFP particles bearing Vpr, 1 µg of Vpr expressing pcDNA3.1 (wild-type SUMA Vpr or Vpr mutants) was transfected with 2.5 µg of pMDG and 2.5 µg of p8.91 in 30 µl Fugene-6 and 500 µl Optimem. All virus supernatants were harvested at 48 and 72 hr post-transfection, replicate flasks were pooled, and supernatants subjected to ultracentrifugation through a 20% sucrose cushion at 23,000 rpm for 2 hr in a 30 ml swingout rotor (Sorvall) (72000G). Viral particles were resuspended in RPMI supplemented with 10% FCS. HIV-GFP produced with p8.91 or p8.2 used in *Figure 1* were DNase treated for 2 hr at 37°C (DNaseI, Sigma) prior to ultracentrifugation. Viruses were titrated by infecting THP-1 cells (2 × $10^5$ cells/ml) with dilutions of sucrose purified virus in the presence of polybrene (8 µg/ml, Sigma) and incubating for 48 hr. GFP-positive, infected cells were counted by flow cytometry using a BD Accuri C6 (BDBiosciences). HIV-1 vector encoding shRNA targeting DCAF1 has been described and was prepared as above (*Berger et al., 2015*).

## SG-PERT

Viral doses were determined by measuring reverse transcriptase activity of virus preparations by qPCR using an SYBR Green-based product-enhanced PCR assay (SG-PERT) as described (*Vermeire et al., 2012*).

## Isolation of primary MDMs and CD4+ T cells from peripheral blood

Primary MDMs were prepared from fresh blood from healthy volunteers. This study was approved by the UCL/UCLH Committees on the Ethics of Human Research, Committee Alpha reference (06/Q0502/92). All participants provided written informed consent and consent for publication. Primary

CD4+ T cells were obtained from leukocyte cones from healthy donors purchased from the National Blood Service UK. Peripheral blood mononuclear cells (PBMCs) were isolated by density gradient centrifugation using Lymphoprep (Stemcell Technologies). For MDM preparation, healthy donor PBMCs were washed three times with PBS and plated to select for adherent cells. Non-adherent cells were washed away after 1.5 hr and the remaining cells incubated in RPMI (Gibco) supplemented with 10% heat-inactivated pooled human serum (Sigma) and 40 ng/ml macrophage colony stimulating factor (R and D systems). Cells were further washed after 3 days and the medium changed to RPMI supplemented with 10% heat-inactivated human serum (Sigma). MDM were then infected 3–4 days later at low multiplicity of infection. Spreading infection was detected by Gag staining and counting Gag-positive cells as described (*Rasaiyaah et al., 2013*). For CD4+ T cells, untouched CD4+ T cells were purified from PBMCs with an indirect magnetic labeling system (MACS, Miltenyi Biotec), according to manufacturer's instructions. Cells were then cultured with 2 µg/ml of plate-bound anti-CD3 and anti-CD28 monoclonal antibodies (αCD3αCD28 stimulation) (mAbs) (eBioscience) and 25 U/ml of recombinant human interleukin-2 (IL-2; Roche Applied Science) at a concentration of $1.5–2 \times 10^6$ cells/ml in RPMI supplemented with 10% heat-inactivated Human Serum (HS) (SigmaAldrich). Cells were maintained at 37°C in 5% $CO_2$ in a humidified incubator for 72 hr. CD4+ T cells were then assessed for spreading infection of CXCR4-tropic HIV-1 NL4.3 WT and ΔVPR at low multiplicity of infection (300 mU of HIV-1 RT Activity per $1 \times 10^6$ cells). Percentage of HIV-1-infected primary CD4+ T cells was determined by flow cytometry measuring p24Gag antigen employing the monoclonal antibody p24Gag-FITC (HIV-1 p24 (24-4), Santa Cruz Biotechnology).

## Innate immune sensing assays

THP-1 cells were seeded in 96-well plates ($5 \times 10^5$ cells/ml). For Vpr expression, cells were infected with an empty or Vpr expressing (pCSVIG) lentiviral vectors for 40 hr. Cell viabilities were similar at 40 hr as assessed by eye, for an example see *Figure 5K*. For stimulation of cells with HT-DNA or poly I:C, 0.2 µl of lipofectamine and 25 µl of Optimem were incubated with HT-DNA or poly I:C (amounts stated in figure legends) for 20 min and added to cells. Lipopolysaccharide (1 µg/ml), TNFα (200 ng/ml), Sendai virus (200 HA U/ml) or cGAMP (5 µg/ml) were added directly to the media. For experiments with virion delivered/associated Vpr, cells were stimulated at the time of infection. Gaussia/Lucia luciferase activities were measured 8 hr post cell stimulation/infection by transferring 10 µl supernatant to a white 96-well assay plate, injecting 50 µl per well of coelenterazine substrate (Nanolight Technologies, 2 µg/ml) and analysing luminescence on a FLUOstar OPTIMA luminometer (Promega). Data were normalized to a mock-treated control to generate a fold induction.

## ELISA

Cell supernatants were harvested for ELISA at 8 hr post-stimulation and stored at −80°C. CXCL-10 protein was measured using Duoset ELISA reagents (R and D Biosystems) according to the manufacturer's instructions.

## ISG qPCR

RNA was extracted from THP-1 cells using a total RNA purification kit (Norgen) according to the manufacturer's protocol. Five hundred ng RNA was used to synthesize cDNA using Superscript III reverse transcriptase (Invitrogen), also according to the manufacturer's protocol. cDNA was diluted 1:5 in water and 2 µl was used as a template for real-time PCR using SYBR Green PCR master mix (Applied Biosystems) and a 7900HT Real-Time PCR machine (Applied Biosystems). Expression of each gene was normalized to an internal control (*GAPDH*) and these values were then normalized to mock-treated control cells to yield a fold induction. The following primers were used:

*GAPDH:* Fwd 5'-GGGAAACTGTGGCGTGAT-3', Rev 5'-GGAGGAGTGGGTGTCGCTGTT-3'
*CXCL-10:* Fwd 5'-TGGCATTCAAGGAGTACCTC-3', Rev 5'-TTGTAGCAATGATCTCAACACG-3'
*IFIT-2:* Fwd 5'-CAGCTGAGAATTGCACTGCAA-3', Rev 5'-CGTAGGCTGCTCTCCAAGGA-3'
*MxA:* Fwd 5'-ATCCTGGGATTTTGGGGCTT-3', Rev 5'-CCGCTTGTCGCTGGTGTCG-3'
*Viperin:* Fwd 5'-CTGTCCGCTGGAAAGTG-3', Rev 5'-GCTTCTTCTACACCAACATCC-3'
*IL-6:* Fwd 5'- AAATTCGGTACATCCTCGACG-3', Rev 5'- GGAAGGTTCAGGTTGTTTTCT-3'

## Immunofluorescence

For confocal microscopy, HeLa cells ($5 \times 10^4$ cells/ml) were seeded into 24-well plates containing sterile glass coverslips. For nuclear translocation assays, we used THP-1 cells ($4 \times 10^5$ cells/ml) adhered in an optical 96-well plate (PerkinElmer) with 50 ng/ml phorbol 12-myristate 13-acetate (PMA, Peprotech) for 48 hr. Where cells were infected and transfected (DNA, PolyI:C) or treated (cGAMP) with innate immune stimulants, the cells were treated or transfected first, and then viral supernatant added to the cultures. Cells were then fixed and stained three hours after this. For fixation, HeLa or adhered THP-1 cells were washed twice with ice-cold PBS and fixed in 4% (vol/vol) paraformaldehyde. Autofluorescence was quenched in 150 mM ammonium chloride, the cells permeabilized in 0.1% (vol/vol) Triton X-100 in PBS and blocked for 30 min in 5% (vol/vol) FCS in PBS. Cells were incubated with primary Ab for 1 hr followed by incubation with secondary Ab for 1 hr. Cells were washed with PBS three times between each step. The coverslips were placed on a slide prepared with a 30 µl drop of mounting medium (Vectashield, containing 4',6-diamidino-2-phenylindole (DAPI)) and allowed to set before storing at 4°C. Images were taken on a Leica TCS SPE confocal microscope and analyzed in ImageJ. For IRF3/NF-κB(p65) translocation, images were taken on Hermes WISCAN (IDEA Bio-Medical) and analyzed with Metamorph software (Molecular Devices). Metamorph calculated a translocation coefficient representing the proportion of staining in nuclear versus cytoplasmic compartments. A value of 1 represents 'all staining in the nucleus', −1 is 'exclusively in cytoplasm' and 0 is 'equally distributed'.

## Immunoblotting

For immunoblotting of viral particles, sucrose purified (as described above) virions ($1 \times 10^{11}$ RT units) were boiled for 10 min in 6X Laemmli buffer (50 mM Tris-HCl (pH 6.8), 2% (w/v) SDS, 10% (v/v) glycerol, 0.1% (w/v) bromophenol blue, 100 mM β-mercaptoethanol) before separating on 12% polyacrylamide gel. Cells were lysed in lysis buffer containing 50 mM Tris pH 8, 150 mM NaCl, 1 mM EDTA, 10% (v/v) glycerol, 1% (v/v) Triton X100, 0.05% (v/v) NP40 supplemented with protease inhibitors (Roche), clarified by centrifugation at 14,000 x *g* for 10 min and boiled in 6X Laemmli buffer for 10 min. Proteins were separated by SDS-PAGE on 12% polyacrylamide gels. Proteins were transferred to a Hybond ECL membrane (Amersham biosciences) using a semi-dry transfer system (Biorad).

## Cell cycle analysis

WT Vpr or Vpr mutants were expressed in THP-1 cells using pCSVIG at an MOI of 1. Cells were incubated for 48 hr and then washed with PBS and fixed in 1 ml cold 70% ethanol on ice for 30 min. To ensure efficient fixing and minimize clumping, ethanol was added dropwise while vortexing. Cell were pelleted in a microfuge and ethanol was removed followed by two wash steps with PBS. To remove RNA from the samples, RNase A (100 µg/ml) was added and the cells were stained with propidium iodide (PI) (50 µ g/ml) to stain cellular DNA. Cells were incubated for 10 min at room temperature and DNA content analyzed by flow cytometry on a BD FACSCalibur (BD Biosciences). The data were analyzed with FlowJo.

## Generation of Vpr mutants

Site directed mutagenesis was performed using Pfu Turbo DNA Polymerase (Agilent) according to the manufacturer's instructions with the following primers using either pCDNA3.1 or pCSVIG encoding SUMA Vpr as template.

> VprF34I+P35N: Fwd 5'-GCCGTGCGGCACATCAACAGACCTTGGCTGCATAGC-3',
> Rev 5'GCTATGCAGCCAAGGTCTGTTGATGTGCCGCACGGC-3'
> VprQ65R: Fwd 5'-GCCATCATCAGAATCCTGCGGCAGCTGCTGTTCATC-3',
> Rev 5'-GATGAACAGCAGCTGCCGCAGGATTCTGATGATGGC-3'
> VprR80A: Fwd 5'-GGCTGCCGGCACAGCGCCATCGGCATCACCCCT-3',
> Rev 5'-AGGGGTGATGCCGATGGCGCTGTGCCGGCAGCC-3'

## Co-immunoprecipitation assays

For KPNA-cargo IPs HEK293T cells were grown in 10 cm dishes and co-transfected with 1 µg of a plasmid expressing FLAG-tagged KPNA1, 1 µg of a plasmid expressing HA-tagged p65 or IRF3 and 1 µg of a plasmid expressing un-tagged SUMA VprF34I+P35N or empty vector control. To account for the effects of SUMA Vpr on expression from CMV promoter-containing plasmids, for IPs containing wild-type SUMA Vpr cells were co-transfected with 2 µg of a plasmid expressing FLAG-tagged KPNA1, 3 µg of a plasmid expressing HA-tagged p65 or IRF3 and 1 µg of a plasmid expressing untagged wild-type SUMA Vpr. All transfection mixes were made up to 6 µg with an empty vector plasmid. After 24 h cells were lysed in lysis buffer (0.5 (v/v)) % NP-40 in PBS supplemented with protease inhibitors (Roche) and phosphatase inhibitors (Roche), pre-cleared by centrifugation and incubated with 25 µl of mouse-anti-HA agarose beads (Millipore) or mouse-anti-FLAG M2 agarose affinity gel (Sigma) for 2–4 hr. Immunoprecipitates were washed three times in 1 ml of lysis buffer and eluted from the beads by boiling in 20 µl of 2X sample buffer containing SDS and β-mercaptoethanol. Proteins were resolved by SDS-polyacrylamide gel electrophoresis (NuPAGE 4–12% Bis-Tris protein gels, Invitrogen) and detected by immunoblotting.

## Statistical analyses

Data were analyzed by statistical tests as indicated in the figure legends. * represent statistical significance: * ($p < 0.05$), ** ($p < 0.01$), *** ($p < 0.001$), **** ($p < 0.0001$).

Representative immunofluorescence images showing IRF3 (red) nuclear translocation in PMA differentiated THP-1 cells treated with cGAMP, or left untreated, and infected with HIV-1 GFP bearing Vpr, or lacking Vpr, or left uninfected. 4′,6-Diamidine-2′-phenylindole dihydrochloride (DAPI) stains nuclear DNA (Blue). Scale bars represent 20 µm.

## Acknowledgements

We thank Veit Hornung for providing THP-1-IFIT-1 cells wild type and knock outs, Neil Perkins for providing NF-KB constructs and advice, Geoffrey Smith for providing constructs encoding KPNA1-3 and Clare Jolly and Richard Sloan for providing NL4.3ΔVpr. This work was funded through an MRC PhD studentship (HK) an MRC Clinical Training Fellowship (CVT), a Wellcome Trust clinical training fellowship (DF), a Wellcome Trust Senior Biomedical Research Fellowship (GJT), the European Research Council under the European Union's Seventh Framework Programme (FP7/2007-2013)/ ERC (grant HIVInnate 339223) (GJT), a Wellcome Trust Collaborative award (GJT) and was supported by the National Institute for Health Research University College London Hospitals Biomedical Research Centre.

## Additional information

### Competing interests

Maria Teresa Rodriguez-Plata: Maria Teresa Rodriguez-Plata is affiliated with Black Belt TX Ltd. The author has no financial interests to declare. The other authors declare that no competing interests exist.

### Funding

| Funder | Grant reference number | Author |
|---|---|---|
| Wellcome Trust | Senior Biomedical Research Fellowship | Greg J Towers |
| H2020 European Research Council | Advanced Grant HIVinnate | Greg J Towers |
| Medical Research Council | PhD studentship | Hataf Khan |
| Medical Research Council | Clinical training fellowship | Chris Van Tulleken |
| Wellcome Trust | Collaborative Award | Greg J Towers |
| National Institute for Health | University College London | Greg J Towers |

| | Hospitals Biomedical Research Centre | |
| --- | --- | --- |
| Wellcome Trust | Clinical Training Fellowship | Douglas Fink |
| European Union Seventh Framework Programme | FP7/2007-2013 | Greg J Towers |
| European Research Council | grant HIVInnate 339223 | Greg J Towers |

The funders had no role in study design, data collection and interpretation, or the decision to submit the work for publication.

## Author contributions

Hataf Khan, Conceptualization, Investigation, Methodology, Writing - original draft, Writing - review and editing; Rebecca P Sumner, Conceptualization, Supervision, Investigation, Methodology, Writing - original draft, Writing - review and editing; Jane Rasaiyaah, Maria Teresa Rodriguez-Plata, Douglas Fink, David Stirling, Investigation, Methodology, Writing - review and editing; Choon Ping Tan, Chris Van Tulleken, Conceptualization, Investigation, Methodology, Writing - review and editing; Lorena Zuliani-Alvarez, Lucy Thorne, Supervision, Investigation, Methodology, Writing - review and editing; Richard SB Milne, Writing - review and editing; Greg J Towers, Conceptualization, Formal analysis, Supervision, Funding acquisition, Writing - original draft, Project administration, Writing - review and editing

## Author ORCIDs

Hataf Khan https://orcid.org/0000-0003-1128-0266
Lorena Zuliani-Alvarez http://orcid.org/0000-0002-4682-4043
Greg J Towers https://orcid.org/0000-0002-7707-0264

## Ethics

Human subjects: This study was approved by the UCL/UCLH Committees on the Ethics of Human Research, Committee Alpha reference (06/Q0502/92). All participants provided written informed consent and consent for publication.

## Decision letter and Author response

Decision letter https://doi.org/10.7554/eLife.60821.sa1
Author response https://doi.org/10.7554/eLife.60821.sa2

# Additional files

## Supplementary files

• Transparent reporting form

## Data availability

All data generated or analysed during this study are included in the manuscript and supporting files.

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
