## [Decision Letter]

**Acceptance summary:**

Vpr is an evolutionarily conserved gene of HIV and SIV, but its role in viral replication is still not clear. In this study, Khan et al. found that Vpr antagonizes innate immune activation by interfering with the nuclear import of two critical immune transcription factors. This mechanism may help unify seemingly disparate Vpr functions spanning viral replication, modulation of nuclear transport, and immune evasion.

**Decision letter after peer review:**

Thank you for submitting your article "HIV-1 Vpr antagonizes innate immune activation by targeting karyopherin-mediated NF- κB/IRF3 nuclear transport" for consideration by *eLife*. Your article has been reviewed by two peer reviewers, and the evaluation has been overseen by a Reviewing Editor and Miles Davenport as the Senior Editor. The reviewers have opted to remain anonymous.

The reviewers have discussed the reviews with one another and the Reviewing Editor has drafted this decision to help you prepare a revised submission.

Summary:

Vpr is an evolutionarily conserved gene of HIV and SIV. Its role in viral replication is still not clear. In this study, Khan et al. found that Vpr antagonizes innate immune activation by interacting with karyopherins and inhibiting IRF3/NF-κB recruitment to KPNA1. This mechanism may help unify disparate Vpr functions spanning viral replication, modulation of nuclear transport, and immune evasion.

Essential revisions:

1) The main statement about Vpr inhibiting NF-κB nuclear translocation is not consistent with previous studies (Liu et al., 2014; Hotter et al., 2017). This issue should be addressed.

2) Recent reports show that HIV-1 infection do not trigger host innate immune activation even in the absence of Vpr (MBio, 2019; PNAS 2020). These results would argue against a role of Vpr in immune suppression.

3) In addition, the mechanism of Vpr antagonizing innate immune responses is quite weak. The data is lack of demonstration of the effect of Vpr on the natural/endogenous KPNA1-IRF3/NF-κB signalosome. Authors should provide sufficient and convincing evidence (ISGs, p-IRF3 and p-p65 expression) of Vpr antagonizing RLR/TLR/STING pathway both in expressed and incorporated manner.

4) The presentation of the history of immunosuppression (which is long and complicated) is helpful but some parts are probably underemphasized. In particular the effects of JEV (Ye et al.) and Hantaan virus (Taylor) and Vaccinia (Pallett) and HCV (Gagne) probably all should be brought up in the Introduction, since they all have to do with NFkB import.

5) Similarly, the published interactions of Vpr with karopherins (Miyatake, Vodicka, Nitahara-Kasahara and maybe more) should be brought up in the Introduction, and this will help make the paper more readable – fewer parts will seem to be coming out of the blue.

6)The summary of history with effects on viral DNA nuclear import here (Discussion paragraph one) is helpful but the authors do not seem to be taking a stance on whether Vpr in fact does promote viral entry into the nucleus (or if they think this was wrong). Some clarity here would be helpful. Also, while targets SLX4 and UNG2 are mentioned it would be helpful to know what their relative importance might be. Or at least what the authors imagine.

7) Similarly, the fact of transmissible founder viruses being largely IFN resistant (Iyer et al.) is mentioned but probably deserves more discussion, with respect to how it relates to Vpr.

8) Finally, the well-established ability of HIV-1 to escape sensing, probably by protection of viral nucleic acids in the capsid, is cited and the activation of IFN seen here is explained by the use of very high MOI. This is probably correct. But the authors do not fully admit to the highly abnormal setting that is needed to induce this activation. The usual lack of sensing, at the more physiologically relevant MOIs, should be viewed as the norm, as shown in the many papers cited, and not dismissed. it should be more strongly emphasized that the induction seen here (and its suppression by Vpr) are happening at artificial circumstances of high MOIs.

Specific experiments/data clarification needed

1) In Figure 1A, 1B, 2A, 2D-F and 3B, the protein levels of ISGs such as IFIT1, IFIT2, MxA in VLP-Vpr infected MDM and THP-1 cells should be shown.

2) In Figure 5, it's interesting that Vpr can suppress p-IRF3, but more data is needed. p-IRF-3 (Ser396) (#4947 Cell Signaling) is commonly used for pIRF3-S396 detection. The protein level of p-IRF-3-S386 needs to be used as a control for Vpr regulating p-IRF-3-S396, together with the same detection methods such as FACS and WB.

3) In Figure 7, low exposure of KPNA1 bands need to be provided in the Co-IP assay in order to prove that the difference in IRF3 or p65 binding ability is not due to the inconsistent expression of KPNA1.

4) More single cell immunofluorescence results of NF-κB nuclear translocation (cGAMP and LPS inducer) and IRF3 nuclear translocation (LPS inducer) are needed to further dissect the role of Vpr.

---

## [Author Response]

Essential revisions:1) The main statement about Vpr inhibiting NF-κB nuclear translocation is not consistent with previous studies (Liu et al., 2014; Hotter et al., 2017). This issue should be addressed.

There are a variety of studies showing inhibition of NF-κB by Vpr and others showing activation. The Hotter paper, which we did not previously cite, is a good example, because it shows both. We have now discussed this issue in the Discussion as follows, citing Hotter’s and Liu’s work.

“Cell type clearly also plays a role in Vpr function. For example, in myeloid cells (Kogan et al., 2013; Miller et al., 2017), and T cells (Ayyavoo et al., 1997), Vpr has been reported to inhibit NF-ĸB. […] We propose that cell type, and the stage of the viral life cycle, influence the effect of Vpr on transcription factor activation.”

2) Recent reports show that HIV-1 infection do not trigger host innate immune activation even in the absence of Vpr (MBio, 2019; PNAS 2020). These results would argue against a role of Vpr in immune suppression.

This is an important point and indeed, there are many studies where HIV infection does not cause an innate immune response, including this one from Cingoz and Goff. Our data show that at lower doses (eg MOI 0.1-0.3 in THP-1 experiments in Figure 1) there is no innate immune response to HIV-1 vector infection. Of course, if there is no response, then Vpr has no impact on innate immune sensing. However, at higher MOI 1-3 sensing is activated and Vpr acts to reduce the inflammatory gene expression changes downstream of sensing. Importantly, in the Cingoz mBio paper, we expect that a failure to activate innate immune sensing is explained by virus dose. Unfortunately, as luciferase viruses were used, the MOI is not apparent and careful comparison of virus dose between studies cannot be made. We have now sought to clarify these points in the Discussion as follows.

“Sensing of HIV-1 is cell type and viral dose dependent. For example, Cingoz et al. reported failure of VSV-G pseudotyped HIV-1 (∆Env, ∆Nef, ∆Vpr) to activate sensing in a variety of cell lines (Cingöz and Goff, 2019). […] This suggests that low MOI infection depends on sensor evasion by viral PAMP sequestration within intact capsids (Jacques et al., 2016) but higher MOI infections can rely on particle associated Vpr to suppress the activation of any exposed viral PAMPs and any endogenous PAMPs that are induced.”

3) In addition, the mechanism of Vpr antagonizing innate immune responses is quite weak. The data is lack of demonstration of the effect of Vpr on the natural/endogenous KPNA1-IRF3/NF-κB signalosome. Authors should provide sufficient and convincing evidence (ISGs, p-IRF3 and p-p65 expression) of Vpr antagonizing RLR/TLR/STING pathway both in expressed and incorporated manner.

We don’t understand what’s being asked here. Almost all of the experiments presented show Vpr mediated suppression of the natural/endogenous innate immune pathways read out by QPCR of host cell gene expression. Some experiments include an IFIT1 reporter, but this is a luciferase gene cloned in place of the natural IFIT1 gene with its natural promoter, normal position within the genome and presumably natural regulation of expression. All luciferase experiments are supported by concordant induction of various endogenous inflammatory genes by qRT-PCR in parallel or in a previous figure. The phrase weak is subjective. In fact, CXCL10 activation by HIV-1 vector is completely suppressed by the presence of virus particle associated Vpr at MOI 1 and almost completely suppressed at MOI 3 (Figure 1C-F). This experiment is using viral particle associated Vpr, delivered by infection, and endogenous innate immune pathways.

The only experiments using exogenously expressed innate immune pathway proteins are the immunoprecipitations demonstrating Vpr-KPNA1 interactions and inhibition of KPNA1 IRF3 and KPNA1-NF-ĸB interactions by wild type but not mutant Vpr. As the reviewers point out, Vpr interaction with KPNA1is already known and now discussed in the Introduction. The point of these experiments in Figure 7 is to use immunoprecipitations to show that Vpr can compete with NF-κB and IRF3 for KPNA1 binding and to show that the defective Vpr mutant Vpr F34I+P35N does not do this, explaining failure to antagonise sensing. These experiments provide mechanistic insight and explain the defective nature of the mutant in antagonising innate immune signals. Competition experiments inevitably require over-expression as its impossible to detect the amount of transcription factor taking part in the experiment by Co-IP when Vpr is delivered asynchronously by viral particles and a natural agonist is used to activate the transcription factors, also asynchronously. Thus, over-expression of the proteins is necessary to study binding specificity and competition, ie between mutant and wild type.

To be honest we’re not really sure what we’re being asked for in this point. The reviewer may not be talking about the IP. But we can’t show competition with particle delivered Vpr and natural activation of eg IRF3.

4) The presentation of the history of immunosuppression (which is long and complicated) is helpful but some parts are probably underemphasized. In particular the effects of JEV (Ye et al.) and Hantaan virus (Taylor) and Vaccinia (Pallett) and HCV (Gagne) probably all should be brought up in the Introduction, since they all have to do with NFkB import.

We’ve moved this text to paragraph three in the Introduction, which now reads:

“Many viruses have been shown to manipulate innate immune activation by targeting transcription factor nuclear entry downstream of PRR. […] Hepatitis C virus NS3/4A protein restricts IRF3 and NF-κB translocation by cleaving KPNB1 (importin-β) (Gagne et al., 2017).“

5) Similarly, the published interactions of Vpr with karopherins (Miyatake, Vodicka, Nitahara-Kasahara and maybe more) should be brought up in the Introduction, and this will help make the paper more readable – fewer parts will seem to be coming out of the blue.

We have begun paragraph four in the Introduction with the explanation that we already knew that Vpr interacted with KPNA1 but that here we link this interaction to inhibition of innate immunity. This paragraph now reads:

“HIV-1 Vpr has also been linked to Karyopherins and manipulation of nuclear import. Vpr has been shown to interact with a variety of mouse (Miyatake et al., 2016), yeast (Vodicka et al., 1998) and human karyopherin proteins including human KPNA1, 2 and 5 (Nitahara-Kasahara et al., 2007). […] Our new findings support a unifying model of Vpr function, consistent with much of the Vpr literature, in which Vpr associated with incoming viral particles suppresses nuclear entry of activated inflammatory transcription factors to facilitate HIV-1 replication in innate immune activated macrophages.”

6)The summary of history with effects on viral DNA nuclear import here (Discussion paragraph one) is helpful but the authors do not seem to be taking a stance on whether Vpr in fact does promote viral entry into the nucleus (or if they think this was wrong). Some clarity here would be helpful. Also, while targets SLX4 and UNG2 are mentioned it would be helpful to know what their relative importance might be. Or at least what the authors imagine.

We were trying to take a stance without being dogmatic. We have now clarified our perspective in the third discussion paragraph and simplified the first discussion paragraph for clarity. The third Discussion paragraph now reads:

“We propose that previous demonstrations of Vpr dependent HIV-1 replication in macrophages, that depended on association of Vpr with the NPC, or with nuclear transport factors, are explained by Vpr inhibition of innate immune sensing and subsequent antiviral responses (Jacquot et al., 2007; Vodicka et al., 1998). […] Concordantly, accidental infection of a lab worker with a Vpr-defective HIV-1 isolate resulted in delayed seroconversion, suppressed viremia and normal T-cell counts without need for anti-viral treatment (Ali et al., 2018).”

With regard to SLX4 and UNG2, in the Discussion, we now say:

“The Vpr R80A mutant, which does not arrest cell cycle, or manipulate SLX4 complex (Gaynor and Chen, 2001; Laguette et al., 2014) was functional in inhibition of innate sensing (Figures 3, 5, 6). Thus we assume that SLX4 interaction does not play a role in innate the immune antagonism shown herein. Mapping the residues of Vpr that are important for innate immune inhibition onto structures resolved by NMR and X-ray crystallography reveals a potentially distinct interface from that targeting UNG2 because residues Vpr 34/35 are distant from the UNG2 binding site (Figure 3—figure supplement 1B, 1C). Further, UNG2 has not been associated with innate immune sensing.”

7) Similarly, the fact of transmissible founder viruses being largely IFN resistant (Iyer et al.) is mentioned but probably deserves more discussion, with respect to how it relates to Vpr.

Iyer’s study doesn’t mention Vpr. Another study, Rossenkhan et al., suggests distinctive amino acid signatures in transmitted founder clone Vprs. This isn’t well understood but we’ve now cited this paper, stating

“A link between escape from innate sensing and successful transmission is suggested by several lines of evidence. These include a generally low HIV transmission frequency (Shaw and Hunter, 2012), the observation that HIV transmitted founder clones are particularly resistant to IFN (Iyer et al., 2017), and encode distinct Vpr amino acid signatures, as compared to chronic viruses (Rossenkhan et al., 2016), as well as the HIV transmission-associated cytokine storm itself (Stacey et al., 2009).”

8) Finally, the well-established ability of HIV-1 to escape sensing, probably by protection of viral nucleic acids in the capsid, is cited and the activation of IFN seen here is explained by the use of very high MOI. This is probably correct. But the authors do not fully admit to the highly abnormal setting that is needed to induce this activation. The usual lack of sensing, at the more physiologically relevant MOIs, should be viewed as the norm, as shown in the many papers cited, and not dismissed. it should be more strongly emphasized that the induction seen here (and its suppression by Vpr) are happening at artificial circumstances of high MOIs.

This is a very important point that we did not deal with in enough detail. It’s overlapping with point 2 above but we’ve cited the new text again in context below. We have now set out our hypothetical model taking into account the data of others in the penultimate discussion paragraph as follows:

“Sensing of HIV-1 is clearly viral dose, and therefore PAMP dose, dependent. For example, Cingoz et al. reported failure of VSV-G pseudotyped HIV-1 (∆Env, ∆Nef, ∆Vpr) to activate sensing in a variety of cell lines (Cingöz and Goff, 2019). […] This suggests that low MOI infection depends on evasion by viral PAMP sequestration within intact capsids (Jacques et al., 2016) but higher MOI infections can rely on particle associated Vpr to suppress the activation of any exposed viral PAMPs and any endogenous PAMPs that are induced.”

Specific experiments/data clarification needed1) In Figure 1A, 1B, 2A, 2D-F and 3B, the protein levels of ISGs such as IFIT1, IFIT2, MxA in VLP-Vpr infected MDM and THP-1 cells should be shown.

This request asks us to repeat a significant proportion of the experiments in the study in order to demonstrate that reduction of mRNA expression equates to reduction of protein expression. Our data already include measurement of secreted CXCL10 protein by ELISA correlating reduction of CXCL10 mRNA expression (Figure 2B), and CXCL10 protein in the cell supernatants (Figure 2C). We have also extensively used the IFIT1 indicator line in which a luciferase gene has been CRISPRed into the IFIT1 locus making luciferase an accurate and effective surrogate for ISG IFIT1 protein expression (Figures 1, 2, 3, 5). Surely we don’t need to redo all these experiments to reiterate a central principal of biology, ie that decreases in mRNA expression cause reduction in protein expression, especially since we have already included examples of protein expression in the manuscript.

2) In Figure 5, it's interesting that Vpr can suppress p-IRF3, but more data is needed. p-IRF-3 (Ser396) (#4947 Cell Signaling) is commonly used for pIRF3-S396 detection. The protein level of p-IRF-3-S386 needs to be used as a control for Vpr regulating p-IRF-3-S396, together with the same detection methods such as FACS and WB.

Unfortunately, the antibody used to detect IRF3 phosphorylation at S396 by flow cytometry (Cell Signaling D6O1M) did not work well in immunoblots in our hands. Further, the antibody detecting IRF3 phosphorylation at S386 by immunoblot (Abcam ab76493) doesn’t work well in flow cytometry, in our hands. The latter point for Abcam anti-S386 is supported by the Abcam website which doesn’t mention flow for this antibody. https://www.abcam.com/irf3-phospho-s386-antibody-epr2346-ab76493.html

Thus we have been unable to use one antibody to control for the phosphorylation using the other. We have of course controlled for IRF3 levels by immunoblot Figure 5 and Figure 5—figure supplement 1 using the Rabbit-anti-IRF3 (from Cell signaling) which is not impacted by phosphorylation. We’re sorry but we don’t think that this request is experimentally tractable.

3) In Figure 7, low exposure of KPNA1 bands need to be provided in the Co-IP assay in order to prove that the difference in IRF3 or p65 binding ability is not due to the inconsistent expression of KPNA1.

A new Figure 7 is provided with reduced exposure to demonstrate similar KPNA1 expression.

4) More single cell immunofluorescence results of NF-κB nuclear translocation (cGAMP and LPS inducer) and IRF3 nuclear translocation (LPS inducer) are needed to further dissect the role of Vpr.

In our hands cGAMP treatment does not activate NF-κB, at least not when measured by nuclear localisation by immunofluorescence. However, we have now included new data demonstrating that Vpr inhibits NF-κB (Figure 6—figure supplement 1C) and IRF3 (Figure 5—figure supplement 1F) nuclear localisation when induced by LPS, as requested.